# PASHA: Efficient HPO and NAS with Progressive Resource Allocation

**Ondrej Bohdal**[1],[*] **Lukas Balles**[2], **Martin Wistuba**[2], **Beyza Ermis**[3],[*]
**Cédric Archambeau**[2], **Giovanni Zappella**[2]
[1]The University of Edinburgh  [2]AWS, Berlin  [3]Cohere for AI
[1]ondrej.bohdal@ed.ac.uk  [3]beyza@cohere.com
[2]{balleslb,marwistu,cedrica,zappella}@amazon.com

## Abstract

Hyperparameter optimization (HPO) and neural architecture search (NAS) are methods of choice to obtain the best-in-class machine learning models, but in practice they can be costly to run. When models are trained on large datasets, tuning them with HPO or NAS rapidly becomes prohibitively expensive for practitioners, even when efficient multi-fidelity methods are employed. We propose an approach to tackle the challenge of tuning machine learning models trained on large datasets with limited computational resources. Our approach, named PASHA, extends ASHA and is able to dynamically allocate maximum resources for the tuning procedure depending on the need. The experimental comparison shows that PASHA identifies well-performing hyperparameter configurations and architectures while consuming significantly fewer computational resources than ASHA.

## 1 Introduction

Hyperparameter optimization (HPO) and neural architecture search (NAS) yield state-of-the-art models, but often are a very costly endeavor, especially when working with large datasets and models. For example, using the results of (Sharir et al., 2020) we can estimate that evaluating 50 configurations for a 340-million-parameter BERT model (Devlin et al., 2019) on the 15GB Wikipedia and Book corpora would cost around $500,000. To make HPO and NAS more efficient, researchers explored how we can learn from cheaper evaluations (e.g. on a subset of the data) to later allocate more resources only to promising configurations. This created a family of methods often described as multi-fidelity methods. Two well-known algorithms in this family are Successive Halving (SH) (Jamieson & Talwalkar, 2016; Karnin et al., 2013) and Hyperband (HB) (Li et al., 2018).

Multi-fidelity methods significantly lower the cost of the tuning. Li et al. (2018) reported speedups up to 30x compared to standard Bayesian Optimization (BO) and up to 70x compared to random search. Unfortunately, the cost of current multi-fidelity methods is still too high for many practitioners, also because of the large datasets used for training the models. As a workaround, they need to design heuristics which can select a set of hyperparameters or an architecture with a cost comparable to training a single configuration, for example, by training the model with multiple configurations for a single epoch and then selecting the best-performing candidate.

On one hand, such heuristics lack robustness and need to be adapted to the specific use-cases in order to provide good results. On the other hand, they build on an extensive amount of practical experience suggesting that multi-fidelity methods are often not sufficiently aggressive in leveraging early performance measurements and that identifying the best performing set of hyperparameters (or the best architecture) does not require training a model until convergence. For example, Bornschein et al. (2020) show that it is possible to find the best hyperparameter – number of channels in ResNet-101 architecture (He et al., 2015) for ImageNet (Deng et al., 2009) – using only one tenth of the data. However, it is not known beforehand that one tenth of data is sufficient for the task.

Our aim is to design a method that consumes fewer resources than standard multi-fidelity algorithms such as Hyperband (Li et al., 2018) or ASHA (Li et al., 2020), and yet is able to identify configurations

---

[*]Work done at AWS, Berlin.

that produce models with a similar predictive performance after full retraining from scratch. Models are commonly retrained on a combination of training and validation sets to obtain the best performance after optimizing the hyperparameters. To achieve the speedup, we propose a variant of ASHA, called Progressive ASHA (PASHA), that starts with a small amount of initial maximum resources and gradually increases them as needed. ASHA in contrast has a fixed amount of maximum resources, which is a hyperparameter defined by the user and is difficult to select. Our empirical evaluation shows PASHA can save a significant amount of resources while finding similarly well-performing configurations as conventional ASHA, reducing the entry barrier to do HPO and NAS.

To summarize, our contributions are as follows: 1) We introduce a new approach called PASHA that dynamically selects the amount of maximum resources to allocate for HPO or NAS (up to a certain budget), 2) Our empirical evaluation shows the approach significantly speeds up HPO and NAS without sacrificing the performance, and 3) We show the approach can be successfully combined with sample-efficient strategies based on Bayesian Optimization, highlighting the generality of our approach. Our implementation is based on the Syne Tune library (Salinas et al., 2022).

## 2 RELATED WORK

Real-world machine learning systems often rely on a large number of hyperparameters and require testing many combinations to identify suitable values. This makes data-inefficient techniques such as Grid Search or Random Search (Bergstra & Bengio, 2012) very expensive in most practical scenarios. Various approaches have been proposed to find good parameters more quickly, and they can be classified into two main families: 1) Bayesian Optimization: evaluates the most promising configurations by modelling their performance. The methods are sample-efficient but often designed for environments with limited amount of parallelism; 2) Multi-fidelity: sequentially allocates more resources to configurations with better performance and allows high level of parallelism during the tuning. Multi-fidelity methods have typically been faster when run at scale and will be the focus of this work. Ideas from these two families can also be combined together, for example as done in BOHB by Falkner et al. (2018), and we will test a similar method in our experiments.

Successive Halving (SH) (Karnin et al., 2013; Jamieson & Talwalkar, 2016) is conceptually the simplest multi-fidelity method. Its key idea is to run all configurations using a small amount of resources and then successively promote only a fraction of the most promising configurations to be trained using more resources. Another popular multi-fidelity method, called Hyperband (Li et al., 2018), performs SH with different early schedules and number of candidate configurations. ASHA (Li et al., 2020) extends the simple and very efficient idea of successive halving by introducing asynchronous evaluation of different configurations, which leads to further practical speedups thanks to better utilisation of workers in a parallel setting.

Related to the problem of efficiency in HPO, cost-aware HPO explicitly accounts for the cost of the evaluations of different configurations. Previous work on cost-aware HPO for multi-fidelity algorithms such as CAHB (Ivkin et al., 2021) keeps a tight control on the budget spent during the HPO process. This is different from our work, as we reduce the budget spent by terminating the HPO procedure early instead of allocating the compute budget in its entirety. Moreover, PASHA could be combined with CAHB to leverage the cost-based resources allocation.

Recently, researchers considered dataset subsampling to speedup HPO and NAS. Shim et al. (2021) have combined coresets with PC-DARTS (Xu et al., 2020) and showed that they can find well-performing architectures using only 10% of the data and 8.8x less search time. Similarly, Visalpara et al. (2021) have combined subset selection methods with the Tree-structured Parzen Estimator (TPE) for HPO (Bergstra et al., 2011). With a 5% subset they obtained between an 8x to 10x speedup compared to standard TPE. However, in both cases it is difficult to say in advance what subsampling ratio to use. For example, the 10% ratio in (Shim et al., 2021) incurs no decrease in accuracy, while reducing further to 2% leads to a substantial (2.6%) drop in accuracy. In practice, it is difficult to find a trade-off between the time required for tuning (proportional to the subset size) and the loss of performance for the final model because these change, sometimes wildly, between datasets. Further, Zhou et al. (2020) have observed that for a fixed number of iterations, rank consistency is better if we use more training samples and fewer epochs rather than fewer training samples and more epochs. This observation gives further motivation for using the whole dataset for HPO/NAS and design new approaches, like PASHA, to save computational resources.

## 3 PROBLEM SETUP

The problem of selecting the best configuration of a machine learning algorithm to be trained is formalized in (Jamieson & Talwalkar, 2016) as a non-stochastic bandit problem. In this setting the learner (the hyperparameter optimizer) receives $N$ hyperparameter configurations and it has to identify the best performing one with the constraint of not spending more than a fixed amount of resources $R$ (e.g. total number of training epochs) on a specific configuration. $R$ is considered given, but in practice users do not have a good way for selecting it, which can have undesirable consequences: if the value is too small, the model performance will be sub-optimal, while if the budget is too large, the user will incur a significant cost without any practical return. This leads users to overestimate $R$, setting it to a large amount of resources in order to guarantee the convergence of the model. We maintain the concept of maximum amount of resources in our algorithm but we prefer to interpret $R$ as a "safety net", a cost not to be surpassed (e.g. in case an error prevents a normal behaviour of the algorithm), instead of the exact amount of resources spent for the optimization.

This setting could be extended with additional assumptions, based on empirical observation, removing some extreme cases and leading to a more practical setup. In particular, when working with large datasets we observe that the curve of the loss for configurations (called arms in the bandit literature) continuously decreases (in expectation). Moreover, "crossing points" between the curves are rare (excluding noise), and they are almost always in the very initial part of the training procedure. Viering & Loog (2021); Mohr & van Rijn (2022) provide an analysis of learning curves and note that in practice most learning curves are well-behaved, with Bornschein et al. (2020); Domhan et al. (2015) reporting similar findings.

More formally, let us define $R$ as the maximum number of resources needed to train an ML algorithm to convergence. Given $\pi_m(i)$ the ranking of configuration $i$ after using $m$ resources for training, there exists minimum $R^*$ much smaller than $R$ such that for all amounts of resources $r$ larger than $R^*$ the rankings of configurations trained with $r$ resources remain the same: $\exists R^* \ll R : \forall i \in \{\text{configurations}\}, \forall r > R^*, \pi_{R^*}(i) = \pi_r(i)$. The existence of such a quantity, limited to the best performing configuration, is also assumed by Jamieson & Talwalkar (2016), and it is leveraged to quantify the budget required to identify the best performing configuration. If we knew $R^*$, it would be sufficient to run all configurations with exactly that amount of resources to identify the best one and then just train the model from scratch with all the data using that configuration. Unfortunately that quantity is unknown and can only be estimated during the optimization procedure. Note that in practice there is noise involved in training of neural networks, so similarly performing configurations will repeatedly swap their ranks.

## 4 METHOD

PASHA is an extension of ASHA (Li et al., 2020) inspired by the "doubling trick" (Auer et al., 1995). PASHA targets improvements for hyperparameter tuning on large datasets by hinging on the assumptions made about the crossing points of the learning curves in Section 3. The algorithm starts with a small initial amount of resources and progressively increases them if the ranking of the configurations in the top two *rungs* (rounds of promotion) has not stabilized. The ability of our approach to stop early automatically is the key benefit. We illustrate the approach in Figure 1, showing how we stop evaluating configurations for additional rungs if rankings are stable.

We describe the details of our proposed approach in Algorithm 1. Given $\eta$, a hyperparameter used both in ASHA and PASHA to control the fraction of configurations to prune, PASHA sets the current maximum resources $R_t$ to be used for evaluating a configuration using the reduction factor $\eta$ and the minimum amount of resources $r$ to be used ($K_t$ is the current maximum rung). The approach increases the maximum number of resources allocated to promising configurations each time the ranking of configurations in the top two rungs becomes inconsistent. For example, if we can currently train configurations up to rung 2 and the ranking of configurations in rung 1 and rung 2 is not consistent, then we allow training part of the configurations up to rung 3, i.e. one additional rung.

The minimum amount of resources $r$ is a hyperparameter to be set by the user. It is significantly easier to set compared to $R$ as $r$ is the minimum amount of resources required to see a meaningful difference in the performance of the models, and it can be easily estimated empirically by running a few small-scale experiments.

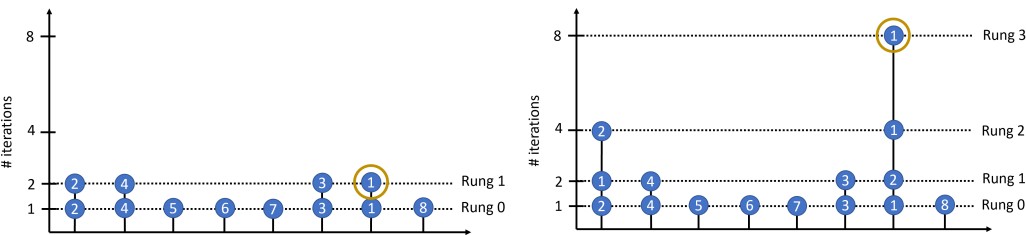

Figure 1: Illustration of how PASHA stops early if the ranking of configurations has stabilized. Left: the ranking of the configurations (displayed inside the circles) has stabilized, so we can select the best configuration and stop the search. Right: the ranking has not stabilized, so we continue.

---

**Algorithm 1** Progressive Asynchronous Successive Halving (PASHA)

1: **input** minimum resource $r$, reduction factor $\eta$

2: **function** PASHA()
3:   $t = 0, R_0 = \eta r, K_0 = \lfloor \log_\eta(R_0/r) \rfloor$
4:   **while** desired **do**
5:     **for** each free worker **do**
6:       $(\theta, k) = \texttt{get\_job}()$
7:       $\texttt{run\_then\_return\_val\_loss}(\theta, r\eta^k)$
8:     **end for**
9:     **for** completed job $(\theta, k)$ with loss $l$ **do**
10:       Update configuration $\theta$ in rung $k$ with loss $l$
11:       **if** $k \geq K_t - 1$ **then**
12:         $\pi_k = \texttt{configuration\_ranking}(k)$
13:       **end if**
14:       **if** $k = K_t$ and $\pi_k \not\equiv \pi_{k-1}$ **then**
15:         $t = t + 1$
16:         $R_t = \eta^t R_0$
17:         $K_t = \lfloor \log_\eta(R_t/r) \rfloor$
18:       **end if**
19:     **end for**
20:   **end while**
21: **end function**

22: **function** get_job()
23:   // Check if there is a promotable config
24:   **for** $k = K_t - 1, \ldots, 1, 0$ **do**
25:     candidates $= \texttt{top\_k}(\text{rung } k, |\text{rung } k|/\eta)$
26:     promotable $= \{c \in \text{candidates} : c \text{ not promoted}\}$
27:     **if** $|\text{promotable}| > 0$ **then**
28:       **return** promotable$[0], k + 1$
29:     **end if**
30:     // If not, grow bottom rung
31:     Draw random configuration $\theta$
32:     **return** $\theta, 0$
33:   **end for**
34: **end function**

---

We also set a maximum amount of resources $R$ so that PASHA can default to ASHA if needed and avoid increasing the resources indefinitely. While it is not generally reached, it provides a safety net.

## 4.1 SOFT RANKING

Due to the noise present in the training process, negligible differences in the measured predictive performance of different configurations can lead to significantly different rankings. For these reasons we adopt what we call "soft ranking". In soft ranking, configurations are still sorted by predictive performance but are considered equivalent if the performance difference is smaller

than a value $\epsilon$ (or equal to it). Instead of producing a sorted list of configuration, this provides a list of lists where for every position of the ranking there is a list of equivalent configurations. The concept is explained graphically in Figure 2, and we also provide a formal definition. For a set of $n$ configurations $c_1, c_2, \cdots, c_i, \cdots, c_n$ and performance metric $f$ (e.g. accuracy) with $f(c_1) \leq f(c_2) \leq \cdots \leq f(c_i) \leq \cdots \leq f(c_n)$, soft rank at position $i$ is defined as

$$\text{soft rank}_i = \{c_j \in \text{configurations} : |f(c_i) - f(c_j)| \leq \epsilon\}.$$

When deciding on if to increase the resources, we go through the ranked list of configurations in the top rung and check if the current configuration at the given rank was in the list of configurations for that rank in the previous rung. If there is a configuration which does not satisfy the condition, we increase resources.

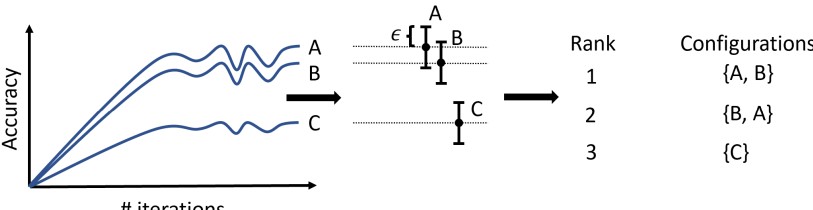

Figure 2: Illustration of soft ranking. There are three lists with the first two containing two items because the scores of the two configurations are closer to each other than $\epsilon$.

## 4.2 AUTOMATIC ESTIMATION OF $\epsilon$ BY MEASURING NOISE IN RANKINGS

Every operation involving randomization gives slightly different results when repeated, the training process and the measurement of performance on the validation set are no exception. In an ideal world, we could repeat the process multiple times to compute empirical mean and variance to make a better decision. Unfortunately this is not possible in our case since the repeating portions of the training process will defeat the purpose of our work: speeding up the tuning process. Understanding when the differences between the performance measured for different configurations are "significant" is crucial for ranking them correctly. We devise a method to estimate a threshold below which differences are meaningless. Our intuition is that configurations with different performance maintain their relative ranking over time. On the other hand, configurations that repeatedly swap their rankings perform similarly well and the performance difference in the current epoch or rung is simply due to noise. We want to measure this noise and use it to automatically estimate the threshold value $\epsilon$ to be used in the soft-ranking described above.

Formally we can define a set of pairs of configurations that perform similarly well by the following:

$$\begin{aligned} S : \{(c, c') : &\left(\pi_{r_j}(c) > \pi_{r_j}(c') \wedge \pi_{r_k}(c) < \pi_{r_k}(c') \wedge \pi_{r_l}(c) > \pi_{r_l}(c')\right) \\ &\vee \left(\pi_{r_j}(c) < \pi_{r_j}(c') \wedge \pi_{r_k}(c) > \pi_{r_k}(c') \wedge \pi_{r_l}(c) < \pi_{r_l}(c')\right)\}, \end{aligned} \tag{1}$$

for resource levels (e.g. epochs – not rungs) $r_j > r_k > r_l$, using the same notation as earlier to refer to resources. In practice we have per-epoch validation performance statistics and use these to find resource levels $r_j, r_k, r_l$ that have configurations with the criss-crossing behaviour (there can be several epochs between such resource levels). We only consider configurations $(c, c')$ that made it to the latest rung, so $r\eta^{K_t - 1} \geq r_j > r\eta^{K_t - 2}$. However, we allow for the criss-crossing to happen across epochs from any rungs. The value of $\epsilon$ can then be calculated as the $N$-th percentile of distances between the performances of configurations in $S$:

$$\epsilon = \mathbf{P}_{N, (c, c') \in S} |f_{r_j}(c) - f_{r_j}(c')|.$$

The exact value of $r_j$ depends on the considered pair of configurations $(c, c')$. To uniquely define $f_{r_j}$, we take the maximum resources $r_j$ currently available for both configurations in the considered pair $(c, c')$. Let us consider the following example setup: the top rung has 8 epochs and

the next one has 4 epochs, there are three configurations $c_a, c_b, c_c$ that made it to the top rung and were trained for 8, 8 and 6 epochs so far respectively. Assuming there was criss-crossing within each pair $(c_a, c_b)$, $(c_a, c_c)$ and $(c_b, c_c)$, the set of distances between configurations in $S$ is $\{|f_8(c_a) - f_8(c_b)|, |f_6(c_a) - f_6(c_c)|, |f_6(c_b) - f_6(c_c)|\}$. The value of $\epsilon$ is recalculated every time we receive new information about the performances of configurations. Initially the value of $\epsilon$ is set to 0, which means that we check for exact ranking if we cannot yet calculate the value of $\epsilon$.

## 5 EXPERIMENTS

In this section we empirically evaluate the performance of PASHA. Its goal is not to provide a model with a higher accuracy, but to identify the best configuration in a shorter amount of time so that we can then re-train the model from scratch. Overall, we target a significantly faster tuning time and on-par predictive performance when comparing with the models identified by state-of-the-art optimizers like ASHA. Re-training after HPO or NAS is important because HPO and NAS in general require to reserve a significant part of the data (often around 20 or 30%) to be used as a validation set. Training with fewer data is not desirable because in practice it is observed that training a model on the union of training and validation sets provides better results.

We tested our method on two different sets of experiments. The first set evaluates the algorithm on NAS problems and uses NASBench201 (Dong & Yang, 2020), while the second set focuses on HPO and was run on two large-scale tasks from PD1 benchmark (Wang et al., 2021).

### 5.1 SETUP

Our experimental setup consists of two phases: 1) run the hyperparameter optimizer until $N = 256$ candidate configurations are evaluated; and 2) use the best configuration identified in the first phase to re-train the model from scratch. For the purpose of these experiments we re-train all the models using only the training set. This avoids introducing an arbitrary choice on the validation set size and allows us to leverage standard benchmarks such as NASBench201. In real-world applications the model can be trained on both training and validation sets. All our results report only the time invested in identifying the best configuration since the re-training time is comparable for all optimizers. All results are averaged over multiple repetitions, with the details specified for each set of experiments separately. We use $N = 90$-th percentile when calculating the value of $\epsilon$.

We use 4 workers to perform parallel and asynchronous evaluations. The choice of $R$ is sensitive for ASHA as it can make the optimizer consume too many resources and penalize the performance. For a fair comparison, we make $R$ dataset-dependent taking the maximum amount of resources in the considered benchmarks. $r$ is also dataset-dependent and $\eta$, the halving factor, is set to 3 unless otherwise specified. The same values are used for both ASHA and PASHA. Runtime reported is the time spent on HPO (without retraining), including the time for computing validation set performance.

We compare PASHA with ASHA (Li et al., 2020), a widely-adopted approach for multi-fidelity HPO, and other relevant baselines. In particular, we consider "one-epoch baseline" that trains all configurations for one epoch (the minimum available resources) and then selects the most promising configuration, and "random baseline" that randomly selects the configuration without any training. For both one-epoch and random baselines we sample $N = 256$ configurations, using the same scheduler and seeds as for PASHA and ASHA. All reported accuracies are after retraining for $R = 200$ epochs. In addition, two, three and five-epoch baselines are evaluated in Appendix A.

### 5.2 NAS EXPERIMENTS

For our NAS experiments we leverage the well-known NASBench201 (Dong & Yang, 2020) benchmark. The task is to identify the network structure providing the best accuracy on three different datasets (CIFAR-10, CIFAR-100 and ImageNet16-120) independently. We use $r = 1$ epoch and $R = 200$ epochs. We repeat the experiments using 5 random seeds for the scheduler and 3 random seeds for NASBench201 (all that are available), resulting in 15 repetitions. Some configurations in NASBench201 do not have all seeds available, so we impute them by averaging over the available seeds. To measure the predictive performance we report the best accuracy on the combined validation and test set provided by the creators of the benchmark.

Table 1: NASBench201 results. PASHA leads to large improvements in runtime, while achieving similar accuracy as ASHA.

| Dataset | Approach | Accuracy (%) | Runtime | Speedup factor | Max resources |
|---|---|---|---|---|---|
| CIFAR-10 | ASHA | 93.85 ± 0.25 | 3.0h ± 0.6h | 1.0x | 200.0 ± 0.0 |
| | PASHA | 93.57 ± 0.75 | 1.3h ± 0.6h | 2.3x | 36.1 ± 50.0 |
| | One-epoch baseline | 93.30 ± 0.61 | 0.3h ± 0.0h | 8.5x | 1.0 ± 0.0 |
| | Random baseline | 72.88 ± 19.20 | 0.0h ± 0.0h | N/A | 0.0 ± 0.0 |
| CIFAR-100 | ASHA | 71.69 ± 1.05 | 3.2h ± 0.9h | 1.0x | 200.0 ± 0.0 |
| | PASHA | 71.84 ± 1.41 | 0.9h ± 0.4h | 3.4x | 20.5 ± 48.3 |
| | One-epoch baseline | 65.57 ± 5.53 | 0.3h ± 0.0h | 9.2x | 1.0 ± 0.0 |
| | Random baseline | 42.83 ± 18.20 | 0.0h ± 0.0h | N/A | 0.0 ± 0.0 |
| ImageNet16-120 | ASHA | 45.63 ± 0.81 | 8.8h ± 2.2h | 1.0x | 200.0 ± 0.0 |
| | PASHA | 45.13 ± 1.51 | 2.9h ± 1.7h | 3.1x | 21.3 ± 48.1 |
| | One-epoch baseline | 41.42 ± 4.98 | 1.0h ± 0.0h | 8.8x | 1.0 ± 0.0 |
| | Random baseline | 20.75 ± 9.97 | 0.0h ± 0.0h | N/A | 0.0 ± 0.0 |

The results in Table 1 suggest PASHA consistently leads to strong improvements in runtime, while achieving similar accuracy values as ASHA. The one-epoch baseline has noticeably worse accuracies than ASHA or PASHA, suggesting that PASHA does a good job of deciding when to continue increasing the resources – it does not stop too early. Random baseline is a lot worse than the one-epoch baseline, so there is value in performing NAS. We also report the maximum resources used to find how early the ranking becomes stable in PASHA. The large variances are caused by stopping HPO at different rung levels for different seeds (e.g. 27 and 81 epochs). Note that the time required to train a model is about 1.3h for CIFAR-10 and CIFAR-100, and about 4.1h for ImageNet16-120, making the total tuning time of PASHA comparable or faster than the training time.

We also ran additional experiments testing PASHA with a reduction factor of $\eta = 2$ and $\eta = 4$ instead of $\eta = 3$, the usage of PASHA as a scheduler in MOBSTER (Klein et al., 2020) and alternative ranking functions. These experiments provided similar findings as the above and are described next.

### 5.2.1 REDUCTION FACTOR

An important parameter for the performance of multi-fidelity algorithms like ASHA is the reduction factor. This hyperparameter controls the fraction of pruned candidates at every rung. The optimal theoretical value is $e$ and it is typically set to 2 or 3. In Table 2 we report the results of the different algorithms ran with $\eta = 2$ and $\eta = 4$ on CIFAR-100 (the full set of results is in Appendix B). The gains are consistent also for $\eta = 2$ and $\eta = 4$, with a larger speedup when using $\eta = 2$ as that allows PASHA to make more decisions and identify earlier that it can stop the search.

Table 2: NASBench201 – CIFAR-100 results with various reduction factors $\eta$. The speedup is large for both $\eta = 2$ and $\eta = 4$, and accuracy similar to ASHA is retained.

| Dataset | Reduction factor | Approach | Accuracy (%) | Runtime | Speedup factor | Max resources |
|---|---|---|---|---|---|---|
| CIFAR-100 | $\eta = 2$ | ASHA | 71.67 ± 0.84 | 3.8h ± 1.0h | 1.0x | 200.0 ± 0.0 |
| | | PASHA | 71.65 ± 1.42 | 0.9h ± 0.1h | 4.2x | 5.9 ± 2.0 |
| | $\eta = 4$ | ASHA | 71.43 ± 1.13 | 2.7h ± 0.9h | 1.0x | 200.0 ± 0.0 |
| | | PASHA | 72.09 ± 1.22 | 1.0h ± 0.4h | 2.8x | 25.1 ± 49.0 |

### 5.2.2 BAYESIAN OPTIMIZATION

Bayesian Optimization combined with multi-fidelity methods such as Successive Halving can improve the predictive performance of the final model (Klein et al., 2020). In this set of experiments, we verify PASHA can speedup also these kinds of methods. Our results are reported in Table 3, where we can clearly see PASHA obtains a similar accuracy result as ASHA with significant speedup.

Table 3: NASBench201 results for ASHA with Bayesian Optimization searcher – MOBSTER (Klein et al., 2020) and similarly extended version of PASHA. The results show PASHA can be successfully combined with a smarter configuration selection strategy.

| Dataset | Approach | Accuracy (%) | Runtime | Speedup factor | Max resources |
|---|---|---|---|---|---|
| CIFAR-10 | MOBSTER | $94.21 \pm 0.28$ | $5.0h \pm 1.1h$ | 1.0x | $200.0 \pm 0.0$ |
| | PASHA BO | $94.00 \pm 0.20$ | $2.6h \pm 1.8h$ | 2.0x | $70.7 \pm 81.6$ |
| CIFAR-100 | MOBSTER | $72.79 \pm 0.68$ | $5.7h \pm 1.4h$ | 1.0x | $200.0 \pm 0.0$ |
| | PASHA BO | $72.16 \pm 1.07$ | $1.6h \pm 0.5h$ | 3.7x | $13.0 \pm 8.7$ |
| ImageNet16-120 | MOBSTER | $46.21 \pm 0.70$ | $15.1h \pm 4.0h$ | 1.0x | $200.0 \pm 0.0$ |
| | PASHA BO | $45.36 \pm 1.06$ | $3.9h \pm 1.2h$ | 3.9x | $11.8 \pm 7.9$ |

### 5.2.3 ALTERNATIVE RANKING FUNCTIONS

We have considered a variety of alternative ranking functions in addition to the soft ranking function that automatically estimates the value of $\epsilon$ by measuring noise in rankings. These include simple ranking (equivalent to soft ranking with $\epsilon = 0.0$), soft ranking with fixed values of $\epsilon$ or obtained using various heuristics (for example based on the standard deviation of objective values in the previous rung), Rank Biased Overlap (RBO) (Webber et al., 2010), and our own reciprocal rank regret metric (RRR) that considers the objective values of configurations. Details of the ranking functions and additional results are in Appendix C.

Table 4 shows a selection of the results on CIFAR-100 with full results in the appendix. We can see there are also other ranking functions that work well and that simple ranking is not sufficiently robust – some benevolence is needed. However, the ranking function that estimates the value of $\epsilon$ by measuring noise in rankings (to which we refer simply as PASHA) remains the easiest to use, is well-motivated and offers both excellent performance and large speedup.

Table 4: NASBench201 – CIFAR-100 results for a variety of ranking functions, showing there are also other well-performing options, even though those are harder to use and are less interpretable.

| Approach | Accuracy (%) | Runtime (s) | Speedup factor | Max resources |
|---|---|---|---|---|
| ASHA | $71.69 \pm 1.05$ | $3.2h \pm 0.9h$ | 1.0x | $200.0 \pm 0.0$ |
| PASHA | $71.84 \pm 1.41$ | $0.9h \pm 0.4h$ | 3.4x | $20.5 \pm 48.3$ |
| PASHA direct ranking | $71.69 \pm 1.05$ | $2.8h \pm 0.7h$ | 1.1x | $200.0 \pm 0.0$ |
| PASHA soft ranking $\epsilon = 2.5\%$ | $71.41 \pm 1.15$ | $1.5h \pm 0.7h$ | 2.1x | $88.3 \pm 74.4$ |
| PASHA soft ranking $\epsilon = 2\sigma$ | $71.14 \pm 0.97$ | $1.9h \pm 0.7h$ | 1.7x | $136.4 \pm 75.8$ |
| PASHA RBO | $71.51 \pm 0.93$ | $2.4h \pm 0.7h$ | 1.3x | $180.5 \pm 50.6$ |
| PASHA RRR | $71.42 \pm 1.51$ | $1.2h \pm 0.5h$ | 2.6x | $39.3 \pm 51.4$ |
| One-epoch baseline | $65.57 \pm 5.53$ | $0.3h \pm 0.0h$ | 9.2x | $1.0 \pm 0.0$ |
| Random baseline | $42.83 \pm 18.20$ | $0.0h \pm 0.0h$ | N/A | $0.0 \pm 0.0$ |

## 5.3 HPO EXPERIMENTS

We further utilize the PD1 HPO benchmark (Wang et al., 2021) to show the usefulness of PASHA in large-scale settings. In particular, we take WMT15 German-English (Bojar et al., 2015) and ImageNet (Deng et al., 2009) datasets that use xformer (Lefaudeux et al., 2021) and ResNet50 (He et al., 2015) models. Both of them are datasets with a large amount of training examples, in particular WMT15 German-English has about 4.5M examples, while ImageNet has about 1.3M examples.

In PD1 we optimize four hyperparameters: base learning rate $\eta \in \left[10^{-5}, 10.0\right]$ (log scale), momentum $1 - \beta \in \left[10^{-3}, 1.0\right]$ (log scale), polynomial learning rate decay schedule power $p \in [0.1, 2.0]$ (linear scale) and decay steps fraction $\lambda \in [0.01, 0.99]$ (linear scale). The minibatch size used for WMT experiments is 64, while the minibatch size for ImageNet experiments is 512. There are 1414 epochs available for WMT and 251 for ImageNet. There are also other datasets in PD1, but these only have a small number of epochs with 1 epoch being the minimum amount of resources. As a result there would not be enough rungs to see benefits of the early stopping provided by PASHA.

If resources could be defined in terms of fractions of epochs, PASHA could be beneficial there too. Most public benchmarks have resources defined in terms of epochs, but in practice it is possible to define resources also in alternative ways. We use 1-NN as a surrogate model for the PD1 benchmark. We repeat our experiments using 5 random seeds and there is only one dataset seed available.

The results in Table 5 show that PASHA leads to large speedups on both WMT and ImageNet datasets. The speedup is particularly impressive for the significantly larger WMT dataset where it is about 15.5x, highlighting how PASHA can significantly accelerate the HPO search on datasets with millions of training examples (WMT has about 4.5M training examples). The one-epoch baseline obtains similar accuracy as ASHA and PASHA for WMT, but performs significantly worse on ImageNet dataset. This result suggests that simple approaches such as the one-epoch baseline are not robust and solutions such as PASHA are needed (which we also saw on NASBench201). Selecting the hyperparameters randomly leads to significantly worse performance than any of the other approaches.

Table 5: Results of the HPO experiments on WMT and ImageNet tasks from the PD1 benchmark. Mean and std of the best validation accuracy (or its equivalent as given in the PD1 benchmark).

| Dataset | Approach | Accuracy (%) | Runtime | Speedup factor | Max resources |
|---|---|---|---|---|---|
| WMT | ASHA | $62.72 \pm 1.41$ | $43.7h \pm 37.2h$ | 1.0x | $1357.4 \pm 80.4$ |
| | PASHA | $62.04 \pm 2.05$ | $2.8h \pm 0.6h$ | 15.5x | $37.8 \pm 21.6$ |
| | One-epoch baseline | $62.36 \pm 1.40$ | $0.6h \pm 0.0h$ | 67.3x | $1.0 \pm 0.0$ |
| | Random baseline | $33.93 \pm 21.96$ | $0.0h \pm 0.0h$ | N/A | $0.0 \pm 0.0$ |
| ImageNet | ASHA | $75.10 \pm 2.03$ | $7.3h \pm 1.2h$ | 1.0x | $251.0 \pm 0.0$ |
| | PASHA | $73.37 \pm 2.71$ | $3.8h \pm 1.0h$ | 1.9x | $45.0 \pm 30.1$ |
| | One-epoch baseline | $63.40 \pm 9.91$ | $1.1h \pm 0.0h$ | 6.7x | $1.0 \pm 0.0$ |
| | Random baseline | $36.94 \pm 31.05$ | $0.0h \pm 0.0h$ | N/A | $0.0 \pm 0.0$ |

## 6 LIMITATIONS

PASHA is designed to speed up finding the best configuration, making HPO and NAS more accessible. To do so, PASHA interrupts the tuning process when it considers the ranking of configurations to be sufficiently stable, not spending resources on evaluating configurations in later rungs. However, the benefits of such mechanism will be small in some circumstances. When the number of rungs is small, there will be few opportunities for PASHA to interrupt the tuning and provide large speedups. This phenomenon is demonstrated in Appendix D on the LCBench benchmark (Zimmer et al., 2021).

Public benchmarks usually fix the minimum resources to one epoch, while the maximum is benchmark-dependent (e.g. 200 epochs for NASBench201 and 50 for LCBench), leaving little control for algorithms like PASHA in some cases. Appendix E analyses the impact of these choices.

For practical usage, we recommend having a maximum amount of resources at least 100 times larger than the minimum amount of resources when using $\eta = 3$ (default). This can be achieved by measuring resources with higher granularity (e.g. in terms of gradient updates) if needed.

## 7 CONCLUSIONS

In this work we have introduced a new variant of Successive Halving called PASHA. Despite its simplicity, PASHA leads to strong improvements in the tuning time. For example, in many cases it reduces the time needed to about one third compared to ASHA without a noticeable impact on the quality of the found configuration. For benchmarks with a small number of rungs (LCBench), PASHA provides more modest speedups but this limitation can be mitigated in practice by adopting a more granular unit of measure for resources. Further work could investigate the definition of rungs and resource levels, with the aim of understanding how they impact the decisions of the algorithm. More broadly this applies not only to PASHA but also to multi-fidelity algorithms in general.

PASHA can also be successfully combined with more advanced search strategies based on Bayesian Optimization to obtain improvements in accuracy at a fraction of the time. In the future, we would like to test combinations of PASHA with transfer-learning techniques for multi-fidelity such as RUSH (Zappella et al., 2021) to further decrease the tuning time.

## REPRODUCIBILITY STATEMENT

We include the code for our approach as part of the supplementary material, including details for how to run the experiments. We use pre-computed benchmarks that make it possible to run the NAS and HPO experiments even without large computational resources. In addition, PASHA is available as part of the Syne Tune library (Salinas et al., 2022).

## ACKNOWLEDGEMENTS

We would like to thank the Syne Tune developers for providing us with a library to easily extend and use in our experiments. We would like to thank Aaron Klein, Matthias Seeger and David Salinas for their support on questions regarding Syne Tune and hyperparameter optimization more broadly. We would also like to thank Valerio Perrone, Sanyam Kapoor and Aditya Rawal for insightful discussions when working on the project. Further, we are thankful to the anonymous reviewers for helping us improve our paper.

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

## A    ADDITIONAL BASELINES

We consider additional baselines that evaluate how good two, three and five-epoch baselines are compared to PASHA. From Table 6 and 7 we see that while these usually get closer to the performance of ASHA and PASHA than the one-epoch baseline, they are still relatively far compared to PASHA. Moreover, it is crucial to observe that such baselines cannot dynamically allocate resources and decide when to stop, and as a result PASHA can outperform them both in terms of speedup and the quality of the found configuration.

Table 6: NASBench201 results. PASHA leads to large improvements in runtime, while achieving similar accuracy as ASHA.

| Dataset | Approach | Accuracy (%) | Runtime | Speedup factor | Max resources |
|---|---|---|---|---|---|
| CIFAR-10 | ASHA | 93.85 ± 0.25 | 3.0h ± 0.6h | 1.0x | 200.0 ± 0.0 |
| | PASHA | 93.57 ± 0.75 | 1.3h ± 0.6h | 2.3x | 36.1 ± 50.0 |
| | One-epoch baseline | 93.30 ± 0.61 | 0.3h ± 0.0h | 8.5x | 1.0 ± 0.0 |
| | Two epoch baseline | 92.75 ± 0.91 | 0.7h ± 0.0h | 4.2x | 2.0 ± 0.0 |
| | Three epoch baseline | 93.47 ± 0.71 | 1.0h ± 0.0h | 2.8x | 3.0 ± 0.0 |
| | Five epoch baseline | 93.38 ± 0.90 | 1.7h ± 0.0h | 1.7x | 5.0 ± 0.0 |
| | Random baseline | 72.88 ± 19.20 | 0.0h ± 0.0h | N/A | 0.0 ± 0.0 |
| CIFAR-100 | ASHA | 71.69 ± 1.05 | 3.2h ± 0.9h | 1.0x | 200.0 ± 0.0 |
| | PASHA | 71.84 ± 1.41 | 0.9h ± 0.4h | 3.4x | 20.5 ± 48.3 |
| | One-epoch baseline | 65.57 ± 5.53 | 0.3h ± 0.0h | 9.2x | 1.0 ± 0.0 |
| | Two-epoch baseline | 68.28 ± 4.25 | 0.7h ± 0.0h | 4.6x | 2.0 ± 0.0 |
| | Three-epoch baseline | 70.47 ± 1.60 | 1.0h ± 0.0h | 3.1x | 3.0 ± 0.0 |
| | Five-epoch baseline | 70.95 ± 0.95 | 1.7h ± 0.0h | 1.8x | 5.0 ± 0.0 |
| | Random baseline | 42.83 ± 18.20 | 0.0h ± 0.0h | N/A | 0.0 ± 0.0 |
| ImageNet16-120 | ASHA | 45.63 ± 0.81 | 8.8h ± 2.2h | 1.0x | 200.0 ± 0.0 |
| | PASHA | 45.13 ± 1.51 | 2.9h ± 1.7h | 3.1x | 21.3 ± 48.1 |
| | One-epoch baseline | 41.42 ± 4.98 | 1.0h ± 0.0h | 8.8x | 1.0 ± 0.0 |
| | Two-epoch baseline | 42.99 ± 1.89 | 2.0h ± 0.0h | 4.4x | 2.0 ± 0.0 |
| | Three-epoch baseline | 44.65 ± 0.95 | 3.0h ± 0.0h | 2.9x | 3.0 ± 0.0 |
| | Five-epoch baseline | 44.75 ± 1.03 | 5.0h ± 0.1h | 1.8x | 5.0 ± 0.0 |
| | Random baseline | 20.75 ± 9.97 | 0.0h ± 0.0h | N/A | 0.0 ± 0.0 |

Table 7:  Results of the HPO experiments on WMT and ImageNet tasks from the PD1 benchmark. Mean and std of the best validation accuracy (or its equivalent as given in the PD1 benchmark).

| Dataset | Approach | Accuracy (%) | Runtime | Speedup factor | Max resources |
|---|---|---|---|---|---|
| WMT | ASHA | 62.72 ± 1.41 | 43.7h ± 37.2h | 1.0x | 1357.4 ± 80.4 |
| | PASHA | 62.04 ± 2.05 | 2.8h ± 0.6h | 15.5x | 37.8 ± 21.6 |
| | One-epoch baseline | 62.36 ± 1.40 | 0.6h ± 0.0h | 67.3x | 1.0 ± 0.0 |
| | Two-epoch baseline | 60.16 ± 3.58 | 1.1h ± 0.0h | 39.1x | 2.0 ± 0.0 |
| | Three-epoch baseline | 61.12 ± 3.47 | 1.6h ± 0.0h | 27.6x | 3.0 ± 0.0 |
| | Five-epoch baseline | 57.89 ± 5.33 | 2.5h ± 0.0h | 17.3x | 5.0 ± 0.0 |
| | Random baseline | 33.93 ± 21.96 | 0.0h ± 0.0h | N/A | 0.0 ± 0.0 |
| ImageNet | ASHA | 75.10 ± 2.03 | 7.3h ± 1.2h | 1.0x | 251.0 ± 0.0 |
| | PASHA | 73.37 ± 2.71 | 3.8h ± 1.0h | 1.9x | 45.0 ± 30.1 |
| | One-epoch baseline | 63.40 ± 9.91 | 1.1h ± 0.0h | 6.7x | 1.0 ± 0.0 |
| | Two-epoch baseline | 64.61 ± 10.81 | 1.7h ± 0.0h | 4.2x | 2.0 ± 0.0 |
| | Three-epoch baseline | 68.74 ± 3.79 | 2.3h ± 0.1h | 3.2x | 3.0 ± 0.0 |
| | Five-epoch baseline | 65.91 ± 3.99 | 3.7h ± 0.1h | 2.0x | 5.0 ± 0.0 |
| | Random baseline | 36.94 ± 31.05 | 0.0h ± 0.0h | N/A | 0.0 ± 0.0 |

## B    REDUCTION FACTOR

Table 8 shows the full set of results for our experiments with different reduction factors. The behaviour is the same across all cases.

Table 8: NASBench201 results with various reduction factors $\eta$.

| Dataset | Reduction factor | Approach | Accuracy (%) | Runtime | Speedup factor | Max resources |
|---------|-----------------|----------|--------------|---------|----------------|---------------|
| CIFAR-10 | $\eta = 2$ | ASHA | $93.88 \pm 0.27$ | $3.6h \pm 1.1h$ | 1.0x | $200.0 \pm 0.0$ |
| | | PASHA | $93.53 \pm 0.76$ | $1.0h \pm 0.3h$ | 3.5x | $9.1 \pm 8.1$ |
| | $\eta = 4$ | ASHA | $93.75 \pm 0.28$ | $2.4h \pm 0.6h$ | 1.0x | $200.0 \pm 0.0$ |
| | | PASHA | $93.65 \pm 0.65$ | $1.1h \pm 0.5h$ | 2.3x | $32.3 \pm 50.2$ |
| CIFAR-100 | $\eta = 2$ | ASHA | $71.67 \pm 0.84$ | $3.8h \pm 1.0h$ | 1.0x | $200.0 \pm 0.0$ |
| | | PASHA | $71.65 \pm 1.42$ | $0.9h \pm 0.1h$ | 4.2x | $5.9 \pm 2.0$ |
| | $\eta = 4$ | ASHA | $71.43 \pm 1.13$ | $2.7h \pm 0.9h$ | 1.0x | $200.0 \pm 0.0$ |
| | | PASHA | $72.09 \pm 1.22$ | $1.0h \pm 0.4h$ | 2.8x | $25.1 \pm 49.0$ |
| ImageNet16-120 | $\eta = 2$ | ASHA | $46.09 \pm 0.68$ | $11.9h \pm 4.0h$ | 1.0x | $200.0 \pm 0.0$ |
| | | PASHA | $45.35 \pm 1.52$ | $2.8h \pm 0.6h$ | 4.2x | $9.3 \pm 7.1$ |
| | $\eta = 4$ | ASHA | $45.43 \pm 0.98$ | $7.9h \pm 3.0h$ | 1.0x | $200.0 \pm 0.0$ |
| | | PASHA | $45.52 \pm 1.30$ | $2.9h \pm 1.1h$ | 2.8x | $18.4 \pm 18.7$ |

## C  EXPERIMENTS WITH ALTERNATIVE RANKING FUNCTIONS

### C.1  DESCRIPTION

PASHA employs a ranking function whose choice is completely arbitrary. In our main set of experiments we used soft ranking that automatically estimates the value of $\epsilon$ by measuring noise in rankings. In this set of experiments we would like to evaluate different criteria to define the ranking of the candidates. We describe the functions considered next.

#### C.1.1  DIRECT RANKING

As a baseline, we study if we can use the simple ranking of configurations by predictive performance (e.g., sorting from the ones with the highest accuracy to the ones with the lowest). If any of the configurations change their order, we consider the ranking unstable and increase the resources.

#### C.1.2  SOFT RANKING VARIATIONS

We consider several variations of soft ranking. The first variation is to fix the value of the $\epsilon$ parameter. We have considered values 0.01, 0.02, 0.025, 0.03, 0.05. The second set of variations aim to estimate the value of $\epsilon$ automatically, using various heuristics. The heuristics we have evaluated include:

- Standard deviation: calculate the standard deviation of the considered performance measure (e.g. accuracy) of the configurations in the previous rung and set a multiple of it as the value of $\epsilon$ – we tried multiples of 1, 2 and 3.
- Mean distance: value of $\epsilon$ is set as the mean distance between the score of the configurations in the previous rung.
- Median distance: similar to the mean distance, but using the median distance.

There are various benefits for estimating the value of $\epsilon$ by measuring noise in rankings, as presented in our paper:

- There is no need to set the value of $\epsilon$ manually.
- Estimation of $\epsilon$ has an intuitive motivation that makes sense.
- The value of $\epsilon$ can dynamically adapt to the different stages of hyperparameter optimization.
- The approach works well in practice.

#### C.1.3  RANK BIASED OVERLAP (RBO) (WEBBER ET AL., 2010)

A score that can be broadly interpreted as a weighted correlation between rankings. We can specify how much we want to prioritize the top of the ranking using parameter $p$ that is between 0.0 and 1.0,

with a smaller value giving more priority to the top of the ranking. The best value is 1.0, while it gives value of 0.0 for rankings that are completely the opposite. We compute the RBO value and then compare it to the selected threshold $t$, increasing the resources if the value is less than the threshold.

### C.1.4  RECIPROCAL RANK REGRET (RRR)

A key insight is that configurations can be very similar to each other and differences in their rankings will not affect the quality of the found solution significantly. To account for this we look at the objective values of the configurations (e.g. accuracy) and compute the relative regret that we would pay at the current rung if we would have assumed the ranking at the previous rung correct.

We define reciprocal rank regret (RRR) as:

$$\text{RRR} = \sum_{i=0}^{n-1} \frac{(f_i - f_i')}{f_i} w^i,$$

where $f$ represents the ordered scores in the top rung, $f'$ represents the reordered scores from the top rung according to the second rung, $n$ is the number of configurations in the top rung and $p$ is the parameter that says how much attention to give to the top of the ranking. The weights $w_i$ sum to 1 and can be selected in different ways to e.g. give more priority to the top of the ranking. For example, we could use the following weights:

$$w_i = \frac{p^i}{\sum_{i=0}^{n-1} p^i}$$

The metric has an intuitive interpretation: it is the average relative regret with priority on top of the ranking. The best value of RRR is 0.0, while the worst possible value is 1.0.

We also consider a version of RRR which considers the absolute values of the differences in the objectives - Absolute RRR (ARRR).

We have evaluated these additional ranking functions using NASBench201 benchmark.

### C.2  RESULTS

We report the results in Table 9, 10 and 11. We see there are also several other variations that achieve strong results across a variety of datasets within NASBench201, most notably soft ranking $2\sigma$ and variations based on RRR. In these cases we obtain similar performance as ASHA, but at a significantly shorter time. We additionally also give a similar analysis in Table 12 (analogous to Table 4), where we analyse a selection of the most interesting ranking functions for the PD1 benchmark.

Table 9: NASBench201 – CIFAR-10 results for a variety of ranking functions.

| Approach | Accuracy (%) | Runtime | Speedup factor | Max resources |
|---|---|---|---|---|
| ASHA | 93.85 ± 0.25 | 3.0h ± 0.6h | 1.0x | 200.0 ± 0.0 |
| PASHA | 93.57 ± 0.75 | 1.3h ± 0.6h | 2.3x | 36.1 ± 50.0 |
| PASHA direct ranking | 93.79 ± 0.26 | 2.7h ± 0.6h | 1.1x | 198.4 ± 6.0 |
| PASHA soft ranking $\epsilon = 0.01$ | 93.79 ± 0.26 | 2.6h ± 0.5h | 1.1x | 194.3 ± 21.2 |
| PASHA soft ranking $\epsilon = 0.02$ | 93.78 ± 0.31 | 2.4h ± 0.5h | 1.2x | 152.4 ± 58.3 |
| PASHA soft ranking $\epsilon = 0.025$ | 93.78 ± 0.31 | 2.3h ± 0.5h | 1.3x | 144.5 ± 59.4 |
| PASHA soft ranking $\epsilon = 0.03$ | 93.78 ± 0.32 | 2.2h ± 0.6h | 1.3x | 128.6 ± 58.3 |
| PASHA soft ranking $\epsilon = 0.05$ | 93.79 ± 0.49 | 1.8h ± 0.7h | 1.6x | 76.0 ± 66.0 |
| PASHA soft ranking $1\sigma$ | 93.75 ± 0.32 | 2.4h ± 0.5h | 1.2x | 186.4 ± 35.2 |
| PASHA soft ranking $2\sigma$ | 93.88 ± 0.28 | 1.9h ± 0.5h | 1.5x | 132.7 ± 68.7 |
| PASHA soft ranking $3\sigma$ | 93.56 ± 0.69 | 0.9h ± 0.3h | 3.1x | 16.2 ± 19.9 |
| PASHA soft ranking mean distance | 93.73 ± 0.52 | 2.3h ± 0.4h | 1.3x | 184.1 ± 40.5 |
| PASHA soft ranking median distance | 93.82 ± 0.26 | 2.3h ± 0.5h | 1.3x | 169.2 ± 51.2 |
| PASHA RBO p=1.0, t=0.5 | 93.49 ± 0.78 | 0.7h ± 0.1h | 4.2x | 4.6 ± 6.0 |
| PASHA RBO p=0.5, t=0.5 | 93.77 ± 0.35 | 2.2h ± 0.6h | 1.3x | 144.0 ± 71.2 |
| PASHA RRR p=1.0, t=0.05 | 93.49 ± 0.78 | 0.7h ± 0.0h | 4.4x | 3.0 ± 0.0 |
| PASHA RRR p=0.5, t=0.05 | 93.76 ± 0.31 | 2.1h ± 0.6h | 1.4x | 140.9 ± 69.7 |
| PASHA ARRR p=1.0, t=0.05 | 93.71 ± 0.35 | 2.4h ± 0.4h | 1.2x | 179.0 ± 42.9 |
| PASHA ARRR p=0.5, t=0.05 | 93.81 ± 0.30 | 2.5h ± 0.4h | 1.2x | 181.0 ± 40.9 |
| One-epoch baseline | 93.30 ± 0.61 | 0.3h ± 0.0h | 8.5x | 1.0 ± 0.0 |
| Random baseline | 72.88 ± 19.20 | 0.0h ± 0.0h | N/A | 0.0 ± 0.0 |

Table 10: NASBench201 – CIFAR-100 results for a variety of ranking functions.

| Approach | Accuracy (%) | Runtime (s) | Speedup factor | Max resources |
|---|---|---|---|---|
| ASHA | 71.69 ± 1.05 | 3.2h ± 0.9h | 1.0x | 200.0 ± 0.0 |
| PASHA | 71.84 ± 1.41 | 0.9h ± 0.4h | 3.4x | 20.5 ± 48.3 |
| PASHA direct ranking | 71.69 ± 1.05 | 2.8h ± 0.7h | 1.1x | 200.0 ± 0.0 |
| PASHA soft ranking $\epsilon = 0.01$ | 71.55 ± 1.04 | 2.5h ± 0.7h | 1.3x | 198.3 ± 6.5 |
| PASHA soft ranking $\epsilon = 0.02$ | 70.94 ± 0.85 | 2.0h ± 0.5h | 1.6x | 160.5 ± 62.9 |
| PASHA soft ranking $\epsilon = 0.025$ | 71.41 ± 1.15 | 1.5h ± 0.7h | 2.1x | 88.3 ± 74.4 |
| PASHA soft ranking $\epsilon = 0.03$ | 71.00 ± 1.38 | 1.0h ± 0.5h | 3.2x | 39.4 ± 63.4 |
| PASHA soft ranking $\epsilon = 0.05$ | 70.71 ± 1.66 | 0.7h ± 0.0h | 4.9x | 3.0 ± 0.0 |
| PASHA soft ranking $1\sigma$ | 71.56 ± 1.03 | 2.5h ± 0.6h | 1.3x | 184.1 ± 40.5 |
| PASHA soft ranking $2\sigma$ | 71.14 ± 0.97 | 1.9h ± 0.7h | 1.7x | 136.4 ± 75.8 |
| PASHA soft ranking $3\sigma$ | 71.63 ± 1.60 | 1.0h ± 0.3h | 3.3x | 20.2 ± 25.3 |
| PASHA soft ranking mean distance | 71.51 ± 0.99 | 2.4h ± 0.5h | 1.4x | 189.8 ± 30.3 |
| PASHA soft ranking median distance | 71.52 ± 0.98 | 2.4h ± 0.6h | 1.3x | 189.5 ± 30.6 |
| PASHA RBO p=1.0, t=0.5 | 70.69 ± 1.67 | 0.7h ± 0.1h | 4.6x | 3.8 ± 2.0 |
| PASHA RBO p=0.5, t=0.5 | 71.51 ± 0.93 | 2.4h ± 0.7h | 1.3x | 180.5 ± 50.6 |
| PASHA RRR p=1.0, t=0.05 | 70.71 ± 1.66 | 0.7h ± 0.0h | 4.9x | 3.0 ± 0.0 |
| PASHA RRR p=0.5, t=0.05 | 71.42 ± 1.51 | 1.2h ± 0.5h | 2.6x | 39.3 ± 51.4 |
| PASHA ARRR p=1.0, t=0.05 | 70.80 ± 1.70 | 0.8h ± 0.4h | 3.8x | 22.9 ± 51.3 |
| PASHA ARRR p=0.5, t=0.05 | 71.41 ± 1.05 | 1.8h ± 0.6h | 1.7x | 110.0 ± 68.7 |
| One-epoch baseline | 65.57 ± 5.53 | 0.3h ± 0.0h | 9.2x | 1.0 ± 0.0 |
| Random baseline | 42.83 ± 18.20 | 0.0h ± 0.0h | N/A | 0.0 ± 0.0 |

Table 11: NASBench201 – ImageNet16-120 results for a variety of ranking functions.

| Approach | Accuracy (%) | Runtime (s) | Speedup factor | Max resources |
|---|---|---|---|---|
| ASHA | 45.63 ± 0.81 | 8.8h ± 2.2h | 1.0x | 200.0 ± 0.0 |
| PASHA | 45.13 ± 1.51 | 2.9h ± 1.7h | 3.1x | 21.3 ± 48.1 |
| PASHA direct ranking | 45.63 ± 0.81 | 8.3h ± 2.5h | 1.1x | 200.0 ± 0.0 |
| PASHA soft ranking $\epsilon = 0.01$ | 45.52 ± 0.89 | 7.0h ± 1.5h | 1.3x | 185.7 ± 36.1 |
| PASHA soft ranking $\epsilon = 0.02$ | 45.79 ± 1.16 | 4.4h ± 1.4h | 2.0x | 71.4 ± 50.8 |
| PASHA soft ranking $\epsilon = 0.025$ | 46.01 ± 1.00 | 3.2h ± 1.0h | 2.8x | 28.6 ± 27.7 |
| PASHA soft ranking $\epsilon = 0.03$ | 45.62 ± 1.48 | 2.4h ± 0.7h | 3.6x | 11.0 ± 10.0 |
| PASHA soft ranking $\epsilon = 0.05$ | 44.90 ± 1.42 | 1.8h ± 0.0h | 5.0x | 3.0 ± 0.0 |
| PASHA soft ranking $1\sigma$ | 45.63 ± 0.89 | 6.5h ± 1.3h | 1.4x | 177.1 ± 44.2 |
| PASHA soft ranking $2\sigma$ | 45.39 ± 1.22 | 4.5h ± 1.4h | 1.9x | 91.2 ± 58.0 |
| PASHA soft ranking $3\sigma$ | 44.90 ± 1.42 | 1.8h ± 0.0h | 5.0x | 3.0 ± 0.0 |
| PASHA soft ranking mean distance | 45.50 ± 1.12 | 6.2h ± 1.5h | 1.4x | 157.7 ± 54.7 |
| PASHA soft ranking median distance | 45.67 ± 0.95 | 6.3h ± 1.6h | 1.4x | 156.3 ± 52.2 |
| PASHA RBO p=1.0, t=0.5 | 44.90 ± 1.42 | 1.8h ± 0.0h | 5.0x | 3.0 ± 0.0 |
| PASHA RBO p=0.5, t=0.5 | 45.24 ± 1.13 | 6.4h ± 1.3h | 1.4x | 148.3 ± 56.9 |
| PASHA RRR p=1.0, t=0.05 | 44.90 ± 1.42 | 1.8h ± 0.0h | 5.0x | 3.0 ± 0.0 |
| PASHA RRR p=0.5, t=0.05 | 44.90 ± 1.42 | 1.8h ± 0.0h | 5.0x | 3.0 ± 0.0 |
| PASHA ARRR p=1.0, t=0.05 | 44.90 ± 1.42 | 1.8h ± 0.0h | 5.0x | 3.0 ± 0.0 |
| PASHA ARRR p=0.5, t=0.05 | 44.90 ± 1.42 | 1.8h ± 0.0h | 5.0x | 3.0 ± 0.0 |
| One-epoch baseline | 41.42 ± 4.98 | 1.0h ± 0.0h | 8.8x | 1.0 ± 0.0 |
| Random baseline | 20.75 ± 9.97 | 0.0h ± 0.0h | N/A | 0.0 ± 0.0 |

Table 12: Results of the HPO experiments on WMT and ImageNet tasks from the PD1 benchmark, using a selection of the most interesting candidates for ranking functions. Mean and std of the best validation accuracy (or its equivalent as given in the PD1 benchmark).

| Dataset | Approach | Accuracy (%) | Runtime | Speedup factor | Max resources |
|---|---|---|---|---|---|
| WMT | ASHA | 62.72 ± 1.41 | 43.7h ± 37.2h | 1.0x | 1357.4 ± 80.4 |
| | PASHA | 62.04 ± 2.05 | 2.8h ± 0.6h | 15.5x | 37.8 ± 21.6 |
| | PASHA direct ranking | 62.16 ± 1.75 | 39.3h ± 38.3h | 1.1x | 1024.0 ± 466.6 |
| | PASHA soft ranking $\epsilon = 2.5\%$ | 62.09 ± 2.04 | 1.3h ± 0.4h | 33.4x | 4.2 ± 2.4 |
| | PASHA soft ranking $2\sigma$ | 62.52 ± 2.18 | 1.1h ± 0.1h | 38.8x | 3.0 ± 0.0 |
| | PASHA RBO p=0.5, t=0.5 | 61.44 ± 1.23 | 6.7h ± 7.8h | 6.5x | 147.6 ± 113.2 |
| | PASHA RRR p=0.5, t=0.05 | 62.52 ± 2.18 | 1.1h ± 0.1h | 38.8x | 3.0 ± 0.0 |
| | One-epoch baseline | 62.36 ± 1.40 | 0.6h ± 0.0h | 67.3x | 1.0 ± 0.0 |
| | Random baseline | 33.93 ± 21.96 | 0.0h ± 0.0h | N/A | 0.0 ± 0.0 |
| ImageNet | ASHA | 75.10 ± 2.03 | 7.3h ± 1.2h | 1.0x | 251.0 ± 0.0 |
| | PASHA | 73.37 ± 2.71 | 3.8h ± 1.0h | 1.9x | 45.0 ± 30.1 |
| | PASHA direct ranking | 75.10 ± 2.03 | 6.8h ± 0.7h | 1.1x | 247.8 ± 3.9 |
| | PASHA soft ranking $\epsilon = 2.5\%$ | 74.73 ± 1.99 | 4.3h ± 2.5h | 1.7x | 140.4 ± 112.8 |
| | PASHA soft ranking $2\sigma$ | 75.82 ± 0.82 | 5.0h ± 1.6h | 1.5x | 133.0 ± 96.8 |
| | PASHA RBO p=0.5, t=0.5 | 74.80 ± 2.19 | 4.4h ± 2.1h | 1.6x | 117.4 ± 109.4 |
| | PASHA RRR p=0.5, t=0.05 | 74.98 ± 2.12 | 1.6h ± 0.0h | 4.7x | 3.0 ± 0.0 |
| | One-epoch baseline | 63.40 ± 9.91 | 1.1h ± 0.0h | 6.7x | 1.0 ± 0.0 |
| | Random baseline | 36.94 ± 31.05 | 0.0h ± 0.0h | N/A | 0.0 ± 0.0 |

# D    ADDITIONAL RESULTS ON LCBENCH

We additionally evaluate PASHA on the LCBench benchmark (Zimmer et al., 2021) where only modest speedups can be expected due to a small number of epochs (and hence rungs) available. LCBench limits the maximum amount of resources per configuration to 50 epochs, so when using and setting the minimum resource level to 1 epoch, it is a challenging testbed for an algorithm like PASHA. The hyperparameters optimized include number of layers $\in [1, 5]$, max. number of units $\in [64, 1024]$ (log scale), batch size $\in [16, 512]$ (log scale), learning rate $\in [10^{-4}, 10^{-1}]$ (log scale), weight decay $\in [10^{-5}, 10^{-1}]$, momentum $\in [0.1, 0.99]$ and max. value of dropout $\in [0.0, 1.0]$. Similarly as in our other experiments, we use $\eta = 3$ and stop after sampling 256 candidates.

Overall, the results in Table 13 confirm an accuracy level on-par with ASHA. While, as expected, the speedup is reduced compared to the experiments on NASBench, in several cases PASHA achieves a 20+% speedup with peaks around 40%.

If only a small number of epochs is sufficient for training the model on the given dataset, then HPO can be performed on a sub-epoch basis, e.g. defining the rung levels in terms of iterations instead of epochs. PASHA would then be able to give a large speedup even in cases with smaller numbers of epochs – an example of which is LCBench.

Table 13: Results of the HPO experiments on the LCBench benchmark. Mean and std of the test accuracy across five random seeds. PASHA achieves similar accuracies as ASHA, but gives only modest speedups because of the limited number of rung levels and opportunities to stop the HPO early. To enable large speedup from PASHA, we could redefine the rung levels in terms of neural network weights updates rather than epochs.

| Dataset | ASHA accuracy (%) | PASHA accuracy (%) | PASHA speedup |
|---|---|---|---|
| APSFailure | $97.52 \pm 0.92$ | $97.01 \pm 0.75$ | 1.3x |
| Amazon_employee_access | $94.01 \pm 0.40$ | $94.21 \pm 0.00$ | 1.1x |
| Australian | $83.35 \pm 0.33$ | $83.35 \pm 0.51$ | 1.1x |
| Fashion-MNIST | $86.70 \pm 1.87$ | $86.34 \pm 1.32$ | 1.1x |
| KDDCup09_appetency | $98.22 \pm 0.00$ | $98.22 \pm 0.00$ | 1.1x |
| MiniBooNE | $86.13 \pm 1.57$ | $86.24 \pm 1.62$ | 1.4x |
| Adult | $79.14 \pm 0.85$ | $79.14 \pm 0.85$ | 1.2x |
| Airlines | $59.57 \pm 1.32$ | $59.22 \pm 0.76$ | 1.4x |
| Albert | $64.31 \pm 0.99$ | $64.23 \pm 0.61$ | 1.2x |
| Bank-marketing | $88.34 \pm 0.07$ | $88.38 \pm 0.00$ | 1.2x |
| Blood-transfusion-service-center | $79.92 \pm 0.20$ | $76.99 \pm 6.00$ | 1.1x |
| Car | $86.60 \pm 6.41$ | $86.60 \pm 6.41$ | 1.1x |
| Christine | $71.05 \pm 1.17$ | $70.15 \pm 1.85$ | 1.2x |
| Cnae-9 | $94.10 \pm 0.31$ | $94.44 \pm 0.11$ | 1.0x |
| Connect-4 | $62.28 \pm 6.87$ | $65.69 \pm 0.04$ | 1.2x |
| Covertype | $59.76 \pm 3.24$ | $61.64 \pm 1.64$ | 1.2x |
| Credit-g | $70.30 \pm 0.84$ | $70.79 \pm 0.68$ | 1.1x |
| Dionis | $64.58 \pm 9.89$ | $64.58 \pm 9.89$ | 1.1x |
| Fabert | $56.11 \pm 10.89$ | $53.47 \pm 9.75$ | 1.1x |
| Helena | $19.16 \pm 3.20$ | $19.16 \pm 3.20$ | 1.1x |
| Higgs | $66.48 \pm 3.16$ | $66.48 \pm 3.16$ | 1.1x |
| Jannis | $58.92 \pm 2.38$ | $59.63 \pm 2.81$ | 1.4x |
| Jasmine | $75.85 \pm 0.36$ | $75.55 \pm 0.68$ | 1.0x |
| Jungle_chess_2pcs_raw_endgame_complete | $72.86 \pm 4.69$ | $74.94 \pm 7.84$ | 1.3x |
| Kc1 | $80.32 \pm 4.37$ | $80.86 \pm 3.37$ | 1.2x |
| Kr-vs-kp | $92.50 \pm 3.93$ | $90.95 \pm 4.70$ | 1.0x |
| Mfeat-factors | $98.21 \pm 0.15$ | $98.15 \pm 0.15$ | 1.1x |
| Nomao | $94.12 \pm 0.60$ | $94.25 \pm 0.64$ | 1.1x |
| Numerai28.6 | $52.03 \pm 0.55$ | $52.30 \pm 0.24$ | 1.3x |
| Phoneme | $76.65 \pm 2.65$ | $75.42 \pm 2.87$ | 1.1x |
| Segment | $83.15 \pm 2.54$ | $83.15 \pm 2.54$ | 1.0x |
| Sylvine | $90.57 \pm 1.87$ | $90.89 \pm 2.04$ | 1.0x |
| Vehicle | $71.76 \pm 2.57$ | $71.76 \pm 2.57$ | 1.1x |
| Volkert | $50.72 \pm 1.91$ | $50.72 \pm 1.91$ | 1.1x |

# E   INVESTIGATION WITH VARIABLE MAXIMUM RESOURCES

We analyse the impact of variable maximum resources (number of epochs) on how large speedup PASHA provides over ASHA. More specifically, we change the maximum resources available for ASHA and also the upper boundary on maximum resources for PASHA. We utilize NASBench201 benchmark for these experiments and set the number of epochs to 200 (default) or 50 (other details are the same as earlier). The results in Table 14 confirm that PASHA leads to larger speedups when there are more epochs (and rung levels) available. This analysis also explains the modest speedups on LCBench analysed earlier.

If the model is trained for a small number of epochs, it is worth redesigning the HPO so that there are more rung levels available, enabling PASHA to give larger speedups. This can be achieved by using sub-epoch resource levels – specifying the rung levels and the minimum resources in terms of the number of iterations (neural network weights updates). Based on the results observed across various benchmarks, we would recommend having at least 5 rung levels in ASHA, with more rung levels leading to larger speedups from PASHA over ASHA.

Table 14: NASBench201 results. PASHA leads to larger speedups if the models are trained with more epochs.

| Dataset | Number of epochs | Approach | Accuracy (%) | Runtime | Speedup factor | Max resources |
|---|---|---|---|---|---|---|
| CIFAR-10 | 200 | ASHA | $93.85 \pm 0.25$ | $3.0h \pm 0.6h$ | 1.0x | $200.0 \pm 0.0$ |
| | | PASHA | $93.57 \pm 0.75$ | $1.3h \pm 0.6h$ | 2.3x | $36.1 \pm 50.0$ |
| | 50 | ASHA | $93.78 \pm 0.39$ | $1.8h \pm 0.2h$ | 1.0x | $50.0 \pm 0.0$ |
| | | PASHA | $93.58 \pm 0.75$ | $1.2h \pm 0.4h$ | 1.5x | $22.0 \pm 16.8$ |
| CIFAR-100 | 200 | ASHA | $71.69 \pm 1.05$ | $3.2h \pm 0.9h$ | 1.0x | $200.0 \pm 0.0$ |
| | | PASHA | $71.84 \pm 1.41$ | $0.9h \pm 0.4h$ | 3.4x | $20.5 \pm 48.3$ |
| | 50 | ASHA | $72.24 \pm 0.87$ | $1.8h \pm 0.3h$ | 1.0x | $50.0 \pm 0.0$ |
| | | PASHA | $71.91 \pm 1.32$ | $0.9h \pm 0.3h$ | 2.0x | $10.5 \pm 12.1$ |
| ImageNet16-120 | 200 | ASHA | $45.63 \pm 0.81$ | $8.8h \pm 2.2h$ | 1.0x | $200.0 \pm 0.0$ |
| | | PASHA | $45.13 \pm 1.51$ | $2.9h \pm 1.7h$ | 3.1x | $21.3 \pm 48.1$ |
| | 50 | ASHA | $45.97 \pm 0.99$ | $5.2h \pm 0.7h$ | 1.0x | $50.0 \pm 0.0$ |
| | | PASHA | $45.09 \pm 1.52$ | $2.7h \pm 1.0h$ | 1.9x | $11.3 \pm 11.7$ |

# F   ANALYSIS OF LEARNING CURVES

We analyse the NASBench201 learning curves in Figure 3 and 4. To make the analysis realistic and easier to grasp, we first sample a random subset of 256 configurations, similarly as we do for our NAS experiments. Figure 3 shows the learning curves of the top three configurations (selected in terms of their final performance). We see that these learning curves are very close to each other and frequently cross due to noise in the training, allowing us to estimate a meaningful value of $\epsilon$ parameter (configurations that repeatedly swap their order are very likely to be similarly good, so we can select any of them because the goal is to find a strong configuration quickly rather than the very best one). Figure 4 shows all learning curves from the same random sample of 256 configurations. In this case we can see that the learning curves are relatively well-behaved (especially the ones at the top), and any exceptions are rare.

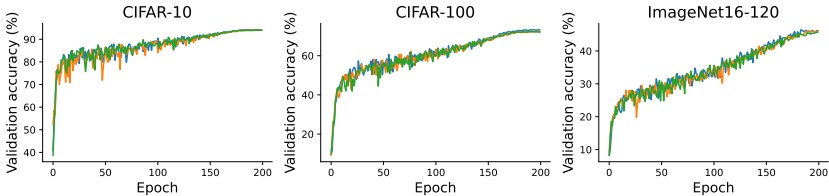

Figure 3: Analysis of how the learning curves of the top three configurations (in terms of final validation accuracy; from a random sample of 256 configurations) evolve across epochs. We see that such similar configurations frequently change their ranks, enabling us to calculate a meaningful value of $\epsilon$ parameter.

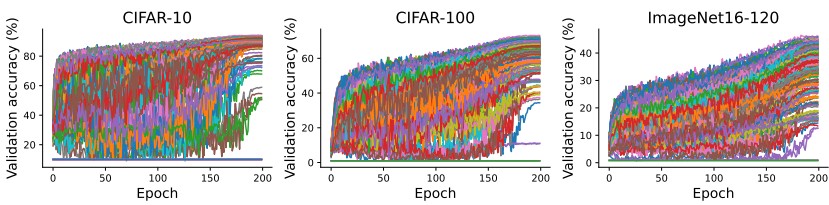

Figure 4: Analysis of what the learning curves look like for a random sample of 256 configurations. We see that the learning curves are relatively well-behaved (especially the ones at the top), and any exceptions are rare.

## G    INVESTIGATION OF HOW VALUE $\epsilon$ EVOLVES

We analyse how the value of $\epsilon$ that is used for calculating soft ranking develops during the HPO process. We show the results in Figure 5 for the three different datasets available in NASBench201 (taking one seed). The results show the obtained values of $\epsilon$ are relatively small.

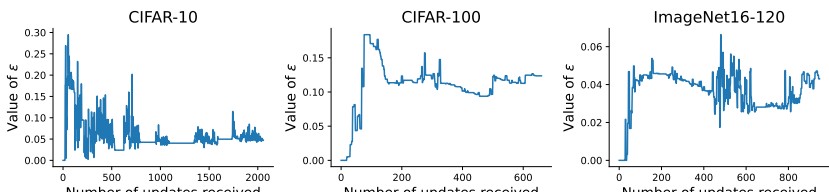

Figure 5: Analysis of how the value of $\epsilon$ evolves as we receive additional updates about the performances of candidate configurations. Note that most of the updates are obtained in the top rung due to how multi-fidelity methods work.

# H INVESTIGATION OF PERCENTILE VALUE $N$

We investigate the impact of using various percentile values $N$ used for estimating the value of $\epsilon$ in Table 15. The intuition is that we want to take some value on the top end rather than the maximum distance in case there are some outliers. We see that the results are relatively stable, even though larger value of $N$ can lead to further speedups. However, from the point of view of a practitioner we would still take $N = 90$ in case there are any outliers in the specific new use-case.

Table 15: NASBench201 results. PASHA leads to large improvements in runtime, while achieving similar accuracy as ASHA. Investigation of various percentile values ($N$) to use for calculating parameter $\epsilon$.

| Dataset | Approach | Accuracy (%) | Runtime | Speedup factor | Max resources |
|---|---|---|---|---|---|
| | ASHA | $93.85 \pm 0.25$ | $3.0h \pm 0.6h$ | 1.0x | $200.0 \pm 0.0$ |
| | PASHA $N = 100\%$ | $93.70 \pm 0.61$ | $1.0h \pm 0.4h$ | 3.0x | $13.8 \pm 19.5$ |
| | PASHA $N = 95\%$ | $93.64 \pm 0.59$ | $1.0h \pm 0.4h$ | 2.8x | $15.4 \pm 19.5$ |
| CIFAR-10 | PASHA $N = 90\%$ | $93.57 \pm 0.75$ | $1.3h \pm 0.6h$ | 2.3x | $36.1 \pm 50.0$ |
| | PASHA $N = 80\%$ | $93.86 \pm 0.53$ | $1.5h \pm 0.6h$ | 1.9x | $60.9 \pm 60.7$ |
| | One-epoch baseline | $93.30 \pm 0.61$ | $0.3h \pm 0.0h$ | 8.5x | $1.0 \pm 0.0$ |
| | Random baseline | $72.88 \pm 19.20$ | $0.0h \pm 0.0h$ | N/A | $0.0 \pm 0.0$ |
| | ASHA | $71.69 \pm 1.05$ | $3.2h \pm 0.9h$ | 1.0x | $200.0 \pm 0.0$ |
| | PASHA $N = 100\%$ | $71.84 \pm 1.41$ | $0.8h \pm 0.1h$ | 3.9x | $6.6 \pm 2.9$ |
| | PASHA $N = 95\%$ | $71.84 \pm 1.41$ | $0.8h \pm 0.1h$ | 3.9x | $6.6 \pm 2.9$ |
| CIFAR-100 | PASHA $N = 90\%$ | $71.91 \pm 1.32$ | $0.9h \pm 0.3h$ | 3.5x | $12.6 \pm 19.2$ |
| | PASHA $N = 80\%$ | $71.78 \pm 1.31$ | $1.2h \pm 0.6h$ | 2.6x | $56.0 \pm 76.2$ |
| | One-epoch baseline | $65.57 \pm 5.53$ | $0.3h \pm 0.0h$ | 9.2x | $1.0 \pm 0.0$ |
| | Random baseline | $42.83 \pm 18.20$ | $0.0h \pm 0.0h$ | N/A | $0.0 \pm 0.0$ |
| | ASHA | $45.63 \pm 0.81$ | $8.8h \pm 2.2h$ | 1.0x | $200.0 \pm 0.0$ |
| | PASHA $N = 100\%$ | $45.09 \pm 1.61$ | $2.3h \pm 0.4h$ | 3.7x | $7.0 \pm 2.8$ |
| | PASHA $N = 95\%$ | $45.26 \pm 1.58$ | $2.4h \pm 0.4h$ | 3.7x | $7.4 \pm 2.7$ |
| ImageNet16-120 | PASHA $N = 90\%$ | $45.13 \pm 1.51$ | $2.9h \pm 1.7h$ | 3.1x | $21.3 \pm 48.1$ |
| | PASHA $N = 80\%$ | $45.36 \pm 1.38$ | $3.6h \pm 1.2h$ | 2.5x | $40.5 \pm 47.7$ |
| | One-epoch baseline | $41.42 \pm 4.98$ | $1.0h \pm 0.0h$ | 8.8x | $1.0 \pm 0.0$ |
| | Random baseline | $20.75 \pm 9.97$ | $0.0h \pm 0.0h$ | N/A | $0.0 \pm 0.0$ |

