# OpenReview forum: "PASHA: Efficient HPO and NAS with Progressive Resource Allocation"
_ICLR.cc/2023/Conference — ICLR 2023 poster_

### Official Review · Reviewer_ZU68 · 2022-10-17

**Confidence:** 3
**Correctness:** 4
**Technical Novelty And Significance:** 3
**Empirical Novelty And Significance:** 3
**Recommendation:** 8

**Clarity, Quality, Novelty And Reproducibility:**

Quality:
The paper is written well. The algorithm is extensively tested and compared to suitable baselines and as such their claims are empirically well-supported.

Clarity:
The paper is generally written clearly, and the figures are useful for understanding the main points. The clarity could be further improved as per my comments in the previous section.

Originality:
The work is novel. While PASHA is an extension of previous work, the core idea of early stopping based on whether rankings have changed between rungs is original to the best of my knowledge.

**Strength And Weaknesses:**

Strengths:
1. The approach is elegant and the primary assumption that learning curves rarely "cross" is supported by previous work.
2. The experiments section is comprehensive and the results are impressive.

Weaknesses:
1. Algorithm 1 is somewhat hard to parse. Some additional comments explaining that $R_t$ and $K_t$ are the current maximum budget and maximum rung might be useful. Also, while Figure 1 is useful for getting the main idea across, a simple sample trajectory of PASHA in the appendix would have been very useful in understanding the finer details of how PASHA handles asynchronicity and growing the bottom rung while still testing higher rungs.

2. Is $N$ (the percentile of distances for the automatic estimation of $\epsilon$) a hyperparameter and if so, should it be an input to Algorithm 1?

**Summary Of The Paper:**

The paper proposes a new HPO algorithm named PASHA. It is an extension of ASHA which was itself an asynchronous extension of Successive Halving. PASHA leverages the idea that learning curves rarely "cross" to derive a novel early stopping rule: PASHA stops allocating resources to configurations at the highest rungs when their relative rankings did not change from the previous rung. They show empirically that PASHA's found best configurations achieve comparative test accuracy to ASHA's while achieving significant speedup.

**Summary Of The Review:**

PASHA's empirical performance is impressive throughout the comprehensive test settings and compared to suitable baselines. The core technical idea is novel and grounded in reasonable assumptions. The paper is well written.

---

> ### Author Response · Authors · 2022-11-16
> **Response**
>
> Thank you very much for the encouraging review, highlighting that the approach is elegant and gives impressive results, and also for providing valuable feedback for our paper!
>
> **Descriptions:** thank you for the suggestion. We now explain in the paper that it is the current maximum budget and maximum rung.
>
> **Value of $N$:** It could be treated as a hyperparameter, but there is no need as we expect this value would be fixed across all cases. It is more to express the intuitive idea that we want to take some value on the top end rather than the maximum distance in case there are some outliers.

---

### Official Review · Reviewer_KgTC · 2022-10-20

**Confidence:** 4
**Correctness:** 3
**Technical Novelty And Significance:** 3
**Empirical Novelty And Significance:** 2
**Recommendation:** 6

**Clarity, Quality, Novelty And Reproducibility:**

###Novelty:

The paper addresses an important and timely problem that was stopping HPO methods from being easily applicable. Nevertheless, it is in fact not the very first paper addressing the problem; but I have to admit that I like the overall idea because of its simplicity and robustness.

### Quality and Clarity:

Overall, the quality of the paper is high and the paper is well written. Nevertheless, I have some doubts and open questions listed below.
In Algorithm1, $R_0$ = $\eta^2r$. Which allows for 3 rungs (r, $\eta r$ and $\eta^2 r$), is that intended? (Only 2 rungs suffice for early-stopping PASHA)

Algorithm 1,  the stopping criterion is missing.

The last paragraph of Section 5.1, “We compare PASHA with ASHA, a state-of-the-art approach for hyperparameter optimization”, should this be “multi-fidelity optimization”?

Given the stopping criterion of PASHA, there is a very important baseline that is missing: training all configurations for 2 epochs. Since PASHA stops early only if the ranking of configurations stabilized in the top 2 rungs, i.e., 2 rungs are the minimal resources that are required by PASHA. Then if we have N configurations, this would take $N \times r $ + $N / \eta \times r \eta$ = $2 \times N r$. Therefore, training all configurations for 2 epochs should be considered as another baseline. (If $R_0$ = $\eta^2r$, then this value should be 3, see the previous comment) For the revision, I would like to ask the authors to add this.

It would be interesting to see if the ranking actually stabilizes if the authors continue to run PASHA until the maximal budget.
Do the authors consider dataset size as another budget type? In their related work, the authors mentioned subset selection and showed that “Further, Zhou et al. have observed that for a fixed number of iterations, rank consistency is better if we use more training samples and fewer epochs rather than fewer training samples and more epochs”

### Reproducibility:

There are several details in the experimental setup that are unclear to me and need clarification. Additionally, code is not provided in the paper.

On Page 5, the authors claim that “Note that resources r_j, r_k, r_l do not need to differ by 1 epoch- there can be e.g. several epochs in between”. However, it is unclear how these values are set in their experiments and unclear the impact of these values.
How do you determine the 90-th percentile of distances to compute the tolerance value $\epsilon$? How does this value influence the performance of PASHA?

Does the random baseline randomly select one configuration from 2560 configurations, if I understand correctly?


**Strength And Weaknesses:**

The authors propose a simple (which is a good characteristic in general) and efficient method based on the assumption that most “cross points” of learning curves occur in the very initial part of the training procedure.  In fact, for users, it is not an easy task to determine the budget steps correctly. So, a contribution to automate this will be highly appreciated and is very timely.

Overall this paper is well-motivated and clearly written. This paper will address the known issue of multi-fidelity optimization: determining the maximal budgets for SHA is hard. Additionally, given the empirical observations of the related work, the authors then propose their own approach in a clear manner and well address the potential issues that appear in their approach.

However, some details are missing or not clear enough for me to understand the paper. Additionally, as this paper presents a heuristic idea, I would like to see a thorough ablation study on PAHSA and a fair comparison with other baselines (see my below comments for details)


**Summary Of The Paper:**

This paper presents a new approach that extends the multi-fidelity optimization approach Asynchronous Successive Halving (ASHA) to Progresive ASHA by iteratively increasing the maximal budgets that are allocated to the candidates configurations. Specifically, the optimization process ends if the ranking of the candidate configurations stabilizes in successive two rungs (rounds of promotion). Experiments show that PASHA can significantly reduce the amount of optimization time while still identifying the optimal configuration


**Summary Of The Review:**

This paper provides a simple yet efficient multi-fidelity optimization approach Progresive ASHA (PASHA) that yields a great speed up while maintaining similar performances when doing multi-fidelity optimization. However, some details are missing or not well supported by the experiments. Therefore, I tend to believe that this paper is not ready to be published at the current stage; maybe nevertheless, if the authors can clarify all of the above points, I would be willing to improve my final score.

---

> ### Author Response · Authors · 2022-11-16
> **Response**
>
> We appreciate the effort and time spent reviewing our paper, and we are grateful that you highlight the simplicity, high usefulness and efficiency of our method. We address the weaknesses and questions next.
>
> * **Initial value of $R_0$:** thanks for spotting this, it should indeed be $R_0=\eta r$ and should only allow for two rungs rather than three. We have updated the pseudocode accordingly and also corrected the other parts of the pseudocode that relate to this. It was only a typo in the pseudocode, the code used to run the experiments is correct.
> * **Stopping criterion:** we specify this in the text, it is the number of sampled configurations (256). We followed the ASHA paper in how we specify the stopping criterion in the algorithm (while “desired”).
> * **Editorial notes:** thanks, we now refer to ASHA as multi-fidelity optimization to make it more precise.
> * **New baselines:** thanks for the suggestion, we have now run these additional baselines. The results are in the supplementary material - Table 7 and 8. While these baselines usually improve over the one-epoch baseline, there is still a substantial gap between these baselines and PASHA. However, the key issue with these additional baselines is that it is impossible to say in advance if 2 or 3 epochs will be enough (these vary even across different runs/random seeds), which is the reason we did not include these baselines earlier. PASHA helps us to estimate a suitable number of epochs automatically.
> * **Stability of rankings:** the rankings never completely stabilise because there is noise involved in training and similarly performing configurations repeatedly swap their rankings.
> * **Dataset size:** we did not consider using it as another budget type, but our method could be used also with such budget types. We would generally recommend using more data for a given number of iterations because prior work has shown this leads to finding better solutions. Note that the existing HPO/NAS benchmarks do not support experiments with dataset size as a resource.
> * **Code:** this is part of supplementary material, as also acknowledged by the other reviewers.
> * **How we calculate resources $r_j, r_k, r_l$** (epochs where order is repeatedly swapped): this is likely to be be more intuitive to understand when expressed as an algorithm (although less concise and less formal):
>     * Given the current progress of HPO, we have information about the per-epoch performance of different configurations. When a new update arrives, we do the following:
>     * For each epoch in the current rung (starting from the latest one), find sets of pairs that have reached the given epoch - we include each pair only once, using the top epoch for which they were trained.
>     * For each considered pair we check if they previously had opposite order to the current one, and even before then if they had the same order as now (i.e. if they repeatedly swapped their ranks) - in any earlier epoch.
>     * If the configurations in the pair repeatedly swapped their ranking, we include the difference between their latest performances in the set S (these performances are from the current rung because we only consider pairs of configurations that reached the current rung).
> * **90th percentile for estimating epsilon:** we used 90% to express the intuitive idea that we want to take some value on the top end rather than the maximum distance in case there are some outliers. We now provide results with various values of $N$ in the appendix (Table 9), and we see that the results are relatively stable, even though larger value of $N$ can lead to further speedups (though from the point of view of a practitioner we would still take $N=90$ in case there are any outliers on the specific considered use-case).
> * **Random baseline:** the goal here is to estimate the performance of a model if we selected its hyperparameters completely randomly, from the whole search space (with no training used to select the hyperparameters). We sample one model at a time, fully train it, and record its performance - repeating this 2560 times. This will give us an estimate of how well a randomly selected model will perform.
>
> We believe we have been able to address all weaknesses and questions, and hope that given the strengths of the paper recognized in this review, the reviewer will recommend acceptance.

---

> > ### Comment · Reviewer_KgTC · 2022-11-23
> > **Increased Score**
> >
> > Hi,
> >
> > Thank you very much for your reply; highly appreciated. It helped me to clarify some misunderstandings. Also in view of reading the other reviews, I share a bit the concern of being incremental and lacking even basic theory, but nevertheless, I also believe that the paper is worth to be published; therefore I increased my score accordingly.

---

> > > ### Author Response · Authors · 2022-11-23
> > > **Thank you**
> > >
> > > Thank you very much for increasing the score, we really appreciate it. We are glad to hear you believe the paper is worth publishing - the concerns regarding lack of basic theory are shared with ASHA and other asynchronous methods that are already widely used, so PASHA is not an exception in this aspect. Even if PASHA could be seen as somewhat incremental with respect to ASHA, the extensions are novel, important and non-trivial, and lead to massive speedups, allowing us to tackle a completely new scale of HPO problems.

---

### Official Review · Reviewer_p4YZ · 2022-10-23

**Confidence:** 3
**Correctness:** 4
**Technical Novelty And Significance:** 3
**Empirical Novelty And Significance:** 2
**Recommendation:** 6

**Clarity, Quality, Novelty And Reproducibility:**

Clarity:
- The paper is overall clear and easy to understand, with the exception mentioned in the weakness section.

Quality:
- The paper is well written, and the visual illustrations (e.g. Figure 2) really help with understanding.

Novelty:
- The proposed algorithm is an improvement over ASHA, yet the modifications seem novel.

Reproducibility:
- Detailed code is provided in the supp file.



**Strength And Weaknesses:**

Strength:

- This paper presents a relevant technique for speeding up multi-fidelity AutoML algorithms.
- The presentation of this paper is clear and straight to the point.
- The proposed algorithm is well motivated, intuitive, simple yet empirically effective.
- Extensive ablations are conducted on the proposed algorithm.

Weakness:

- Clarity:
    - While the beginning of section 4 is crystal clear, I find the narrative of Section 4.1 looks a bit detached. It seems to be the first time that “soft-ranking” has been mentioned in the paper. So I would expect some motivations for it at the beginning rather than going straight into explaining its computation.
- Empirical results:
    - While the run-time reduction is significant, in several cases PASHA produces worse final accuracy than ASHA. For instance, Table 1 CIFAR-10 and TinyImageNet, and all of Table 3 and 5. I wonder what would be the run-time reduction if the authors aligned the performance with ASHA?
- Evaluation metrics:
    - For the experimental section, the author mainly uses the final performance of the selected configurations as the evaluation metric (additionally regret in the appendix). For ranking-based search algorithms, it might also be worthwhile to evaluate the global and top-K ranking correlations. I am aware that the soft-ranking technique would make it a bit tricky to compute these metrics. But maybe you can simply switch to hard ranking when computing this metric. I feel the proposed method could produce a better or comparable correlation than ASHA or SHA.

**Summary Of The Paper:**

This paper studies multi-fidelity AutoML algorithms and proposes an improvement over ASHA.

Inspired by the observation that there might exist a cross-over point of the learning curves of different configurations where their ranking swaps, the author propose to leverage this instability of ranking to progressively allocate more resources. Further, a soft-ranking technique is proposed for smoothing out potential noise, with a threshold parameter that can be automatically decided rather than tuned.

Empirical evaluations are conducted on NAS-Bench-201 and PD1 HPO Bench. Compared with ASHA, the proposed method achieves a 2-10 times reduction in run-time while sometimes falling short of the final performance.

**Summary Of The Review:**

I find the presented method intuitive and relevant. My main concern is with experimental results, i.e. while PASHA successfully reduces the run-time, it often also performs worse than ASHA as well. It would be great if the author could also assess the run-time reduction by matching the final performance with ASHA on these datasets and benchmarks.

---

> ### Author Response · Authors · 2022-11-16
> **Response**
>
> We appreciate the encouraging review and valuable feedback, and are grateful for highlighting that we present a well motivated, intuitive, simple yet empirically effective technique. We address the questions from the review next:
>
> * **Clarity - soft ranking introduction:** that is a great suggestion, we now briefly introduce and motivate soft ranking in section 3.
> * **Empirical results:** the key detail is that even if the final accuracy is sometimes worse compared to ASHA, it is still close to the accuracy of the configuration selected by ASHA. The accuracies of other fast approaches such as one-epoch selection or selecting the configuration completely randomly are a lot worse, proving that PASHA can indeed find strong configurations. Ensuring that we always find the same configuration as ASHA would require us to not do early stopping in ASHA because there is some level of noise in the ranking of configurations, making it impossible to always recover the same configuration while using fewer epochs. We show that PASHA manages to find an excellent trade-off between HPO speed and accuracy of the selected configuration.
> * **Alternative evaluation metrics:** thank you for this idea, we have already tried using ranking correlations and they work relatively well (see the supplementary material, Section F). However, these introduce hard to set hyperparameters such as the threshold value. We wanted to ensure that the way we do early stopping is intuitive and without hyperparameters that are hard to set.

---

### Official Review · Reviewer_oH7E · 2022-11-03

**Confidence:** 3
**Correctness:** 4
**Technical Novelty And Significance:** 2
**Empirical Novelty And Significance:** 3
**Recommendation:** 6

**Clarity, Quality, Novelty And Reproducibility:**

**Clarity:**
The paper is well-organized and clearly written.

**Quality:**
The paper appears to be technically sound. The experimental evaluation is adequate, and the results convincingly support the main claims.

**Novelty:**
The paper contributes some new ideas or represents incremental advances.

**Reproducibility:**
The code is available, and the experimental setup is comprehensively described. Any competent researchers can easily reproduce the main results.

**Details Of Ethics Concerns:**

I do not find any ethical concerns.

**Strength And Weaknesses:**

**Strengths of paper:**
1. The problem studied in the paper is interesting and has many real-life applications, as some of them (HPO and NAS) are mentioned in this paper.

2. The authors empirically show that their proposed algorithm speeds up on HPO and NAS datasets while only having minimal performance degradation.


**Weaknesses of paper:**
1. Main weakness of the paper is its novelty, as it is an extension of existing work (algorithm ASHA).

2. The word 'efficient' is used in the title, which signifies that their algorithm achieves the best performance. However, apart from empirical results, there is no principled way to say that their algorithms in indeed efficient.

3. No theoretical guarantee: There is no theoretical guarantee of how far (in terms of accuracy) the final model is after using PASHA from the best possible model.

4. It is unclear how to choose the value of $\epsilon$ principled way. This choice is important as choosing very small or large $\epsilon$ adversely affects the performance.

**Question and other comments.**

Please address the above weakness. I have a few more questions/comments:
1. Can PASHA exploit the estimated function values (like BO) to choose the next configurations?
2. In conclusion: Mention the dataset name instead 'the largest dataset with millions of training examples.'

I am open to changing my score based on the authors' responses.

**Summary Of The Paper:**

This paper studies the problem of efficient resource allocation by identifying the well-performing configurations for hyperparameter optimization and neural architecture search. The goal is to find the best (or close to the best) configurations while using as few computational resources as possible, leading to a faster algorithm.


The authors propose an algorithm named PASHA (Adaptive ASHA), a variant of the existing state-of-the-art algorithm ASHA for the same problem. Compared to a fixed amount of maximum resources for each iteration in ASHA, PASHA initially starts with a small amount of maximum resources and then gradually increases as needed. The authors have empirically validated that the PASHA is faster than ASHA while incurring a minimal penalty on the final model's accuracy on HPO and NAS datasets.

**Summary Of The Review:**

This paper has some overlap with my current work. My recent work was focused on closely related topics and I am knowledgeable about most of the topics covered by the paper.

---

> ### Author Response · Authors · 2022-11-16
> **Response**
>
> We appreciate the effort spent reviewing our paper and recognizing the practical impact of our work, as well as highlighting the speedups of our method while retaining most of the accuracy. We address the weaknesses and questions next.
>
> * **Novelty:** even though PASHA extends ASHA, we believe the modifications are non-trivial and make our method reasonably strong on novelty, as also recognized by other reviewers (key novelties include how to enable early stopping in ASHA and how to do it automatically in a principled way). Further, both ASHA and Hyperband are extensions of Successive Halving, yet they were published in top venues and are widely regarded as novel algorithms by the community. PASHA leads to large speedups over ASHA and makes it possible to apply HPO to qualitatively different settings (HPO on massive datasets), so the impact is far from incremental (e.g. not like 1-2% improvement in accuracy that many papers do) - we enable ML practitioners to scale HPO to massive datasets and save them a lot of compute.
> * **Efficiency:** it is the design of our algorithm that makes it possible to achieve significant speedups and achieve efficiency. The results show PASHA has significantly better speed compared to ASHA, yet achieves similar accuracy, which means that it can far more efficiently find a strong solution. Of course, it could theoretically happen that there is an HPO problem where it is necessary to perform full ASHA to find a strong configuration, but our approach has a mechanism to automatically decide this and based on our experimental investigation it appears such cases are rare.
> * **Theoretical guarantees:** To the best of our knowledge commonly used methods such as ASHA do not offer any theoretical guarantee, this is probably due to the complexity of analysing an asynchronous algorithm of this kind. Theoretical guarantees are available for Successive Halving (SH) – the synchronous version of ASHA – but this requires significantly more time for the optimization. Moreover, the guarantees provided in the non-stochastic case for SH are mostly based on assumptions guaranteeing a large-enough budget. Budget has a significantly different interpretation in the asynchronous case to the point that it is not even mentioned in our pseudocode. We probably could provide an analysis for a synchronous version of PASHA but that will probably end up being irrelevant for practical purposes given the differences introduced by the asynchronous case.
> * **How to choose the value of epsilon in a principled way:** we have a dedicated section in the paper (section 4.2) that explains how to find the value of epsilon in a principled yet simple way. The approach is based on measuring what epsilon is equivalent to noise in the rankings, and it is one of our main contributions that makes PASHA especially novel.
> * **Can PASHA exploit the estimated function values (like BO) to choose the next configurations?** Yes, PASHA is able to exploit BO searchers to sample the next configurations, and we show this in section 5.2.2. We compare there with MOBSTER, which is equivalent to ASHA with BO search strategy.
> * **Editorial notes:** thanks, we now mention in the conclusion that it is the WMT dataset that has millions of training examples.
>
> We believe we have been able to address all weaknesses and questions, and we hope that given the strengths of the paper recognized in this review, the reviewer will recommend acceptance.

---

> > ### Comment · Reviewer_oH7E · 2022-11-25
> > **Increased score**
> >
> > Hi,
> >
> > Thank you very much for your detailed responses to address my concerns! After reading other reviews and your responses, I am increasing my score.

---

> > > ### Author Response · Authors · 2022-11-25
> > > **Thank you**
> > >
> > > Thank you very much for increasing the score, we really appreciate it. We are glad to hear our responses have addressed your concerns.

---

### Official Review · Reviewer_tjkb · 2022-11-04

**Confidence:** 4
**Correctness:** 3
**Technical Novelty And Significance:** 2
**Empirical Novelty And Significance:** 3
**Recommendation:** 6

**Clarity, Quality, Novelty And Reproducibility:**

The writing is overall clear. Sections 4.2 and 6 stood out as sore thumbs to me given the rest was a smooth read. Some details on the experiments could be elaborated more.

The motivation of the work is well founded and an important problem in making HPO more accessible to the community and lowering down costs required for practical HPO. The paper identifies a possible direction for it and presents it well. A popular algorithm is taken as the baseline and is appended with a feature to address a potential flaw in it. The motivation, setup, and presentation of the work is a classic example of empirical research and does well at that. However, the content and presentation of the empirical data could be improved and more convincing.

**Details Of Ethics Concerns:**

No ethical concerns. Adds a feature to an algorithm used widely in various tools.

**Strength And Weaknesses:**

Strengths:

* Relevant and well-placed motivation and problem statement.
* Takes an existing, popular algorithm to potentially improve it and lower computational costs in achieving similar results.
* Simplicity of the approach and seemingly easy-to-implement changes to ASHA.
* Relaxes the need for one of the 2 important ASHA hyperparameters (assuming  $\eta=3$ is standard).
* Incorporates the possibility of LCs crisscrossing, unlike vanilla ASHA.
* Implementation integrated into an existing library SyneTune.

Weaknesses:

A) Incoherency stems when reading more of the paper.
  1. It is not clear what *training from scratch* means on Page 1 given the setup of the problem. When the algorithm proposed is designed where the user is not expected to design or choose the max resource $R$, does training from scratch implies training till convergence? This means the training can be till $<R$, $=R$ or  $>R$. And thus it seemed that the tables reported are for this number.
  2. Similarly, for Random Search (RS) as mentioned at the end of Section 5.1, it is not clear what *look up* means in the cost of a benchmark where a learning curve can be queried anywhere. One would presume, RS is the performance of a configuration at the $R$ available for NAS-201. If so, the table is further confusing. Given RS uses the same seeds as PASHA and ASHA and samples 10x more times, it should recover the configs found by PASHA and ASHA. Both of these will likely select different incumbents. When reporting the performance of training from scratch, the incumbent performance is taken to be the best till convergence (*epochs* $\in[r, R]$). However, RS reports the performance for those incumbents at a (potentially) higher epoch $R$. It is a bit confusing.
  3. The introduction highlights the example of one of the large language models from 2019 and its associated cost. We have much larger models and potentially more expensive ones. However, the strong assumptions made about the shape of LCs seem to be from literature not to do with this problem scope. It is not that the citations are misplaced but rather the expectation set for the reader was a bit different in the beginning. However, this relates to the only Transformer based benchmarks in the experiments in Section 5.3, and PASHA is underwhelming there.

B) Strong assumptions that go into the design of the algorithm and the experiments.
  1. The existence of crossing points only at the beginning of training is a strong one. Given the subtleties of incumbent selection and promotion of configurations involved. The literature cited as reference points to LCs over data subsets and other well-behaved parametric forms of learning curves.
  2. One argument is that ASHA doesn't even consider this so PASHA is doing better there. However, Hyperband's sampling at different rungs was shown to be crucial in handling crisscrossing LCs. One could cite that as a possible reason for Mobster being better than PASHA in terms of performance.
  3. Though PASHA could argue that it prevents unnecessary evaluations at higher fidelities/rungs, there is not enough motivation to not be sampling directly at the rungs already allowed for (L32 in Algorithm 1 returns the base rung).
  4. This is especially important to understand given that PASHA makes assumptions on the performance correlation across rungs in its dynamic heuristic. Especially looking at the RS numbers in the experiments, one wonders what the shapes of the LCs are in the benchmarks as they suggest heavy divergence with more budget. This intuitively may not be the case for the recent large models.
  5. Section 4.2 is entirely dependent on this assumption and is one of the key points in the paper. Looking at the conjunction in the definition of set $S$, at least 3 rungs are required for PASHA to work and find config pairs with crisscrossing. The inequalities over the disjunction suggest that a pair of configurations need to criss-cross at least twice. This seems to be linked to the earlier assumption, ''configurations that repeatedly swap their rankings...''. Again, a fair but strong assumption on which to base the primary contribution. Empirical evidence to support this is also weak. For example, visualizing LCs from the tabular benchmarks could be a great start to untangling this.
  6. For large architectural spaces, the LCs could offer different convergence rates and might have just one crossing point. Based on the notation, this pairing will not be considered as part of noise calculation. Which sounds completely fine. However, quite clear that one of the improving LCs can never be promoted. Given PASHA has a view over this already by feature, one wonders if PASHA could tackle it. One defence for PASHA could be that look, ASHA too will fail here.

C) Uclear or points which raise questions.
  1. If 3 rungs are at least required for PASHA to be effective, does that mean the user should always go with the lowest possible minimum resource better than random, such that enough rungs can be found with the $log_{\eta}(R/r)$ calculation? If so, this must be called out more clearly.
  2. When set $S$ is constructed, pairs are collected based on the 3 highest rungs. After that, the noise $\epsilon$ is calculated based only on the performance difference at the highest rung only. Soft ranking across rungs use this same $\epsilon$. Given that the previous rung is $\eta$ iterations earlier, I wonder if the noise estimated at different epochs/rungs should be done per rung. That is, not sure if the soft-ranking computation using $\epsilon$ from noise at a higher rung is applicable to the lower rung. Also, given the earlier assumption of crisscrossing, isn't the noise in the higher rung to be lesser?
  3. This further intrigues the evolution of $\epsilon$ over a PASHA run. Especially to relate to the ablation with fixed $\epsilon$. That is, does $\epsilon$ shrink to zero as we go higher and thus naturally switch to ASHA? Or does the $\epsilon$ stabilize and PASHA thus converges to a fixed ranking and never add a rung?
  4. It is not at all clear how the incumbents are selected if no high-budget rungs are ever added. Since the top rank can be shared by multiple configurations, as they differ by the current $\epsilon$. However, it seems that the incumbent $x^*$ is selected as the top-1 at a rung, going by L25 of Algorithm 1. Given the assumptions, not sure if this is the right thing to report and could potentially explain the underperforming of PASHA in many experiments.
  5. This further raises the question of somehow the soft ranking should be used for the top-k calculation itself when selecting for promotions. If PASHA trusts the noise measure to declare that all performances within the $\epsilon$ bound are similar so it is safe to stop early, it is counter-intuitive to do an $argmax$ within that $\epsilon$ bound.
  6. In Section 5.2 for the NAS-201 experiments, it is said that the predictive performance over both the validation+test sets are reported. This is confusing given the earlier problem setup of retraining over the full training set (training+validation). My current belief is that the incumbent was chosen over the performance of the validation set alone. The chosen incumbent's performance over both valid+test is shown in the table. However, I am not at all sure.
  7. In Section 5.2 there's an impressive statement there regarding PASHA's total running time, which if true, should be highlighted more. However, what do the model training times mentioned there (1.3h & 4.1h) mean? Training a model for $R$ or *till convergence which could be $>R$. Not clear to me.

D) One clear pathology that is not explored.
  1. As I understand, PASHA will create 3 rungs to enable the noise calculation and soft ranking. There might be this one case that the ranking in the top-2 rungs never changes. In that case, I don't see how PASHA creates a new higher rung. I wonder if the low max resources in PASHA, reported in the tables, have anything to do with this. Pair that with the cases with comparable performances of 1-epoch baselines, it is tough to pinpoint if PASHA avoids this pathology. Or if this pathology is something to have when tackling a general, unknown HPO problem.

Other comments:

0. The Tables are difficult to parse, especially the ablations. Some guides to the reader could be provided with bolds, underlines, colours, etc.
1. In Related Work, it is said that ASHA does better resource allocation than SH. I think given the use of *resource* in the paper, it should be called out that ASHA showed a better utilisation of *workers* in a parallel setting. In a synchronous single-worker case, it can be expected that SH's wider rung selections do better than ASHA in the beginning.
2. Page 5, for example in the last paragraph might have a typo. If not, it was quite confusing to follow. If the top rung has 8 epochs and the previous rung had 4 (more likely than 6 with $\eta=\{2,3,4\}$), it is said that config $c_c$ is at 6 epochs. That is one source of confusion. Secondly, the set of distances considered is across the rungs too. Whereas, in the previous paras it is said that only $(c, c')$ which have made it to the last rung are considered in the set $S$.
3. Parsing of all results has been a bit of a swing from understanding to not being sure and back and forth, simply due to the lack of clarity I have on the evaluation protocol. Even for the one-epoch baseline, are we evaluating at $R$ for the table? Or is it the performances at 1-epoch? Since RS must recover this same configuration and evaluate at $R$. Does then the tables suggest that the selection procedure at 1-epoch is more reliable than $R$? Again, from the current paper draft, this is difficult to disentangle.
4. Looking at the performance tables, it would be interesting to see ASHA's performances and speedups given the max resources a run of PASHA explores. That would create different rungs and budget spacing for ASHA but would still give an insight into what provides the occasional gains for PASHA. Especially when linked with the evolution of $\epsilon$. Is then the main question (quite rightly) the allocation of budgets and the overall budget for exploration at lower rungs?
5. Two points that come up from the previous statement
    * Can we perform Hyperband in $[r, R*]$ and sample at the explored rungs directly? Can that do better than PASHA? How might that affect $\epsilon$ calculation over 3 rungs?
    * Given that PASHA offers speed-ups, can it switch or slowly move to ASHA to allow for higher rung evaluations? A dynamic heuristic that could respond to unchanging rankings over time, or to a width limit for a rung (inspired by Hyperband) could avoid the pathology mentioned earlier too. While not affecting the overall speed gains over ASHA.
6. Mobster results on PD1 would be good to see.
7. Elucidating on the choice of 1-NN surrogate for PD1 should be explained more along with the distance measure used for. Given that log-scaled parameters exists and the deep learning HPO landscape could be irregular, a cross-validated surrogate performance analysis (or equivalent) should be a good addition to the appendix.
8. Ablation studies on PD1, especially for Table 7 would be nice to have.
9. Table 7 is fascinating but also raises the question as to why any of the other ranking methods were not selected. There are also methods which are comparable to PASHA and offer more speedups using even lesser max resources. This again begs two more questions:
    * How much of this is a contribution of the noise band calculation with respect to the learning curves being optimized?
    * What are the learning curves like? (And what is the evaluation protocol for ASHA, PASHA, and baselines)

**Summary Of The Paper:**

The paper identifies the problem of high practical costs even in parallelizable, efficient, multi-fidelity hyperparameter optimization (HPO) methods when it comes to modern, large, deep learning models. The primary contribution is a modification of the popular Asynchronous Successive Halving (ASHA) method that can dynamically choose if a configuration should be evaluated longer to keep making decisions on the best-found configuration.

The proposed algorithm, Progressive-ASHA (PASHA), is a rather simple and intuitive extension to ASHA where a noise measure is dynamically tracked based on the set of partial learning curves seen that criss-cross each other. A soft ranking measure is used for the tournament in the ASHA rungs, where multiple configurations can share a rank if their performances at the highest recorded epoch is within the noise calculated. If only such rankings are ever changed in the top two consecutive rungs, a new higher rung is added based on ASHA. Thus, the maximum resource ($R$) hyperparameter to ASHA is not a strict requirement for PASHA and only is the upper bound of epochs for any configuration. The method appears to offer impressive speedups over ASHA. Not necessarily in superior performance, but generally reaches ballpark performance much quicker. Therefore, PASHA could be seen as an alternative to ASHA to be quicker, or be seen as an improvement to ASHA altogether.


**Summary Of The Review:**

The paper is a nicely motivated, well-written work of a simple and intuitive idea that was put together effectively in working code. The scope is one of an important need to the community and on practical grounds, the algorithm PASHA offers itself as a potential candidate to be used in place of ASHA. Thus, in practice, there is evidence to suggest that PASHA can be a viable, practical and efficient option. However, in terms of a manuscript that researchers can cite to base newer algorithms on PASHA, just as PASHA does with ASHA, it falls short. The experiment setup is not 100\% clear and coherent after a few passes of the paper (have not seen the code). The empirical evidence thus feels inadequate. Moreover, given the strong assumptions on learning curves that PASHA is based on, the entire experiment setup, benchmarks, LCs are all very black-box and don't help convince the use of PASHA as a general HPO algorithm for an unknown task.

---

> ### Author Response · Authors · 2022-11-16
> **Response (1/6)**
>
> Thank you very much for the in-depth review with valuable feedback and advice. We appreciate the significant effort spent reviewing our paper and recognizing the many strengths of our approach . We believe many of the comments listed in weaknesses (or other parts) stem from two key misunderstandings - the correct version is as follows:
> * Our random baseline is not random search, it is a strategy selecting a random configuration from the search space.
> * Estimation of epsilon value uses per-epoch performance observations rather than per-rung observations.
>
> We have updated our paper with clarifications, and we explain more details and the significant impact of these misunderstandings when addressing the specific points raised in the review.

---

> ### Author Response · Authors · 2022-11-16
> **Response (2/6) - Section A - Incoherency**
>
> A-1) **Training from scratch:** once we find the hyperparameters, we fully retrain the model until convergence. Max resource $R$ is typically the number of epochs needed to reach convergence, and the final accuracies in the tables are for this number. The motivation to do retraining is that validation set helps even if the dataset is large - practitioners usually find hyperparameters and then train on all data, that is the use-case we tackle. We do not do that in our experiments to be able to use standard public benchmarks and to make our results comparable with others in the literature. Reproducibility is important for our research and this choice does not affect the benefit in real-world applications where additional data can be used.
>
> A-2) **Random search:** our “Random” baseline is **not** random search. Our random baseline samples a configuration randomly from the whole search space, but these are not trained for selecting the hyperparameters. It is a simple baseline that helps quantifying the benefit of hyperparameters tuning in general and avoids making claims on results obtained on degenerate search spaces. As part of the approach, we sample one model at a time, fully train it, and record its performance - repeating this 2560 times. This will give us an estimate of how well a randomly selected model will perform.
>
> A-3) **LLMs in introduction:** we tried to give a concrete example that shows large-scale HPO can be very expensive. Given the available literature on this topic, it is the best we could find and is a trade-off between trying to give a specific example for motivation and making it related to our work. Regarding the statement that PASHA is underwhelming for WMT: even if the one epoch baseline happens to give a good result there, this approach is hard to generalize (e.g., see results with 2/3/5 epochs). PASHA gives an impressive ~15x speedup on WMT compared to ASHA, while retaining similar accuracy - so we believe PASHA gives excellent results on WMT. Moreover, we adopted the definition of resources as epochs to be able to leverage pre-computed public benchmarks, but it is possible to define the resources with sub-epoch granularity (e.g., 1 unit of resources = 100 gradient updates) and probably obtain even bigger speedup benefits.

---

> ### Author Response · Authors · 2022-11-16
> **Response (3/6) - Section B - Assumptions**
>
> B-1/B-2) Crossing points: this is an assumption used by the whole family of multi-fidelity methods (including successive halving) and so is fair to assume since these methods are widely used in practice. We also ground our assumptions based on several earlier papers (Domhan et al., IJCAI’15; Viering and Loog, arXiv’21; Mohr and van Rijn, arXiv’22), so the assumptions are grounded in existing literature. The empirical results in previous work  suggest this is a reasonable assumption to make (Jamieson et al., AISTATS’16; Li et al., MLSys’20).
>
> Similarly, works on learning curve forecasting make similar assumptions:
> Swersky et al.: Freeze-thaw Bayesian optimization
> Domhan et al.: Speeding up automatic hyperparameter optimization of deep neural networks by extrapolation of learning curves
> Klein et al.: Learning Curve Prediction with Bayesian Neural Networks
>
> We now also include additional analysis of learning curves in the supplementary material to give further support (Figure 4 and 5).
>
> B-3) L32 in the algorithm means that when we sample a new configuration, we initially only train it with the minimum resources $r$, same as for ASHA. This makes sense to do because ASHA (and PASHA) does the first round of pruning at the initial rung.
>
> B-4) We believe this point stems from confusing our random baseline with random search.
>
> B-5,6 and C-1,2) These questions and comments stem from the misunderstanding of how we estimate the value of epsilon. As we mention in the paper, the computations in section 4.2 are defined in terms of epochs rather than rung levels. We need at least 3 epochs to see the criss-crossing, and not 3 rungs (2 rungs already include 3 epochs). We now analyse the learning curves in the supplementary material and show that similarly performing configurations frequently cross each other, allowing us to estimate a meaningful value of epsilon (Figure 4). With a large number of candidates, even in a large search space some of the sampled configurations would be similar and because of that they would repeatedly swap their ranking (noise in training of neural networks). To make it more intuitive how the epsilon is calculated, we included a description of how this is implemented as an algorithm in response for reviewer KgTC - How we calculate resources $r_j, r_k, r_l$. We’ve also made small updates to the text in the paper.

---

> ### Author Response · Authors · 2022-11-16
> **Response (4/6) - Section C - Points which raise questions**
>
> C-1/2) See part "B-5,6 and C-1,2)" in Response (3/6)
>
> C-3) Value of epsilon stabilises at a non-zero value, and this is expected because there is noise in training of neural networks - the performance goes up and down a little across various epochs.
>
> C-4) High-budget rungs are added if the soft ranking becomes unstable. If the soft ranking is stable, we select the best performing configuration in the latest rung. The key part is that this accuracy is still similar to that obtained by ASHA, with the well-performing configuration being found much faster.
>
> C-5) The case of pruning requiring splitting of configurations in the same position is not very likely. If that happens, then we will have to break the tie in some way and we decide to break it with argmax because that will select one or more among a set of equivalent configurations. It is not counterintuitive.
>
> C-6) We make the distinction between academic benchmarks and how the algorithm would be used in practice. In practice, you would retrain on the whole combination of training and validation set for best performance. However, this is not how the academic benchmarks such as NAS-201 are constructed - we have to adjust to how the benchmark was constructed in order to be able to use public standard benchmarks.
>
> C-7) We retrain until $R=200$ epochs, which is the maximum available value in NAS-201. It also corresponds to the number of epochs that would be typically needed for convergence. Note that we don't include the retraining time in the calculation because it would add noise (different configurations have different cost) and it would not make a difference since none of the algorithms are aware of the configuration cost. For NAS-201 experiments, the retraining time is similar to the time needed by PASHA to find a strong configuration, which we believe highlights the impressive speedups obtained by PASHA.

---

> ### Author Response · Authors · 2022-11-16
> **Response (5/6) - Section D and other comments**
>
> **Section D:**
>
> D-1) The pathology stems from earlier misunderstanding. We updated the text and clarified the situation earlier. Please let us know if you have more questions on this topic.
>
> **Other comments:**
>
> OC-0) Thanks for the suggestion, we have tried to make it easier by explaining in the caption what exactly to notice in the table. Note that highlighting specific parts of the table would not be useful as it is the combination between speed and accuracy that is interesting.
>
> OC-1) Thanks for the correction, we have updated the text accordingly.
>
> OC-2) This example tries to make it clear that noise estimation is not on the rung level but on epoch level.
>
> OC-3) Yes, the accuracies are after full retraining at $R=200$, as implied by the first paragraph in subsection 5.1. However, we have improved the text to clarify this. Further confusion in this comment stems from misunderstanding our random baseline for random search.
>
> OC-4) That will require additional work in our codebase and we did not have time to carry out this experiment so far because of the extensive amount of results added to answer the other questions. However, since in that case ASHA’s behavior will closely resemble the behavior of PASHA, we expect the result to be similar.
>
> OC-5) Extending PASHA to be a Hyperband-like (or ASHA with multiple brackets if you prefer) is definitely possible. In our preliminary experiments we did not see significant differences in accuracy between single-bracket ASHA and multi-bracket ASHA, but we saw the multi-bracket being slower. So we focused on extending ASHA, and that remains the main priority for us given the practical impact. A dynamically shrinking epsilon-value is an interesting idea, but it is hard to predict how that will work in practice.
>
> OC-6) Unfortunately MOBSTER takes too much time for ASHA in the case of PD1 benchmark, so as of now we have not been able to obtain the result (also because of the time spent on the results mentioned above). We will do our best to add more results later.
>
> OC-7) We have selected the 1-NN surrogate as it is the recommended option in Syne Tune library. This is also the one used in Salinas et al., AutoML’22.
>
> OC-8) Thanks for the suggestion of including an ablation study of different ranking functions for PD1 (Table 13). We now include it in the supplementary material, in a section designed for evaluating the different ranking functions.
>
> OC-9) We designed PASHA to be a good trade-off between performance, automation and simplicity. The main reason the other ranking functions were not selected is that they require hard to set hyperparameters, and also that they are not as simple as the selected version. As there was also the recommendation to include analysis of learning curves, we include these too in the supplementary material.

---

> ### Author Response · Authors · 2022-11-16
> **Response (6/6) - Summary**
>
> **Clarity:** Section 4.2 was likely unclear because of the incorrect expectation that it uses noise across rungs rather than epochs. We tried to make sure it is clear that epochs are used by clearly mentioning it in the text and also giving a specific example. There were no specific comments about lack of clarity in section 6 in this review, so we are not sure what the reviewer meant.
>
> **Summary - LC assumptions:** these assumptions are grounded in literature as we have explained earlier in answer to B-1, and common algorithms such as ASHA or Successive Halving rely on them too.  Hence we believe the assumptions are perfectly reasonable.
>
> **Summary - black box nature:** We hope that after clarifying the situations and explaining the misunderstandings regarding random search and the estimation of epsilon this comment can be considered outdated. We believe the algorithm is not black-box at all because the inner workings of the algorithm can be inspected and are easily interpretable thanks to the intuitive meaning of the epsilon parameter. Please let us know if you have additional questions on this point.
>
> Overall we believe we have been able to address the weaknesses and questions raised in the review, and we hope the reviewer is willing to recommend acceptance for the paper.

---

> > ### Comment · Reviewer_tjkb · 2022-11-20
> > **Enumerated reponse to the rebuttal**
> >
> > Responses indexed as per the indexes used in the author's response:
> >
> > A-1) What I intended to clarify was if the performance reported is y(x,R) or max(y(x,r), y(x,r+1), ..., y(x,R)). Though that is not explicitly clear, given the revision, I understand it to be the former.
> >
> > A-2) Thanks for clarifying. In that case, I feel the baseline should also sample just N=256, using the same seeds. It feels like a better indication of the quality of random samples that ASHA/PASHA can select in its 256 samples. The quality of the search space could be a separate analysis.
> >
> > A-3) Thank you for the 2-3-5 epoch baselines. They are interesting. This indicates that 5-epoch baselines already offer good prediction quality at $R$ for the benchmark.
> >
> > B-5,6 and C-1,2) In the example at the end of Sec 4.2, the two high rungs are mapped to epochs 4 and 8 and $c_{c}$ is going to be evaluated till 8 epochs. And therefore, $c_{c}$ is considered in set $S$? The clarification of $r_{\{i,j,k\}}$ as epochs and rungs do help. Does then the example given imply that $c_{\{a,b,c\}}$ were checked for crisscrossing on epochs $4,5,6$ (as per Eq. 1)?
> >
> > C-5) If at the highest rung, more than one configurations share the same soft rank, how reliable does an $argmax$ measure become if $R^*$ is still in a small range? Given benchmarks are being used, if the mean, std. dev of the performances at $R$ of these configs sharing the top rank is seen, does the std dev fall in the difference of performance with ASHA?
> >
> > D-1) I am not sure I see the changes regarding this. I may not have been clear and if I understand correctly, PASHA can still just reach a state wherein the rankings don't change at all and a higher rung is never added. This appears to be by design ("PASHA finds the well-performing configuration much faster"). My question/concern is do we want this? If we continue to spend budget on evaluating more configurations, the trade-off in going for a higher rung for a more robust and reliable estimate.
> >
> > OC-5) My intuition of the decay stems from how one could go towards ASHA and gain stronger estimates. As I alluded to earlier, if PASHA already gains amazing performance in a shorter time, IF, PASHA is run for longer, for the equivalent budget to ASHA, then can it explore higher rungs and promote much stronger candidates than ASHA? Thereby, making much better decisions on higher resource allocation. In that case, PASHA could likely be beating ASHA consistently. In that way, PASHA won't appear to be a plugin feature to ASHA but a new algorithm that is faster and potentially better than ASHA.
> >
> > OC-6) No problem, and understandable. When one does PASHA with GPs, I believe the GP is fit over the fidelity as one extra dimension (like Mobster). For the acquisition function, at which epoch is the performance estimated?
> >
> > OC-7) Fair enough. Not saying it is a necessary requirement to change the score, however, it could be good practice to include some analysis in the appendix, showing the surrogate quality.
> >
> > Fig 3) Is the y-axis on the scale of accuracy? Does that mean that the final converged epsilon is around 5% accuracy? And the x-axis implies the number of updates to state information, i.e., every epoch of a configuration.
> >
> > Fig 4) I wonder how to read this presentation. Firstly, might it be better to visualize LCs only till the max resource PASHA explores to get a better idea of the noise and the potential epsilon, therefore? Secondly, based on the paper text, and method design, the $argmax$ over the top 3 LCs at $R^*$ will be the incumbent selected. Given that these seem to crisscross a lot, all these 3 should share the same rank or be considered *similar*. Does this plot intend to corroborate that?
> >
> > Regarding Section 6)
> > Firstly, I believe that by design *other* multi-fidelity algorithms such as Hyperband or async Hyperband have been designed to try and avoid the situation of early pruning a potential good configuration. Albeit at the cost of more resources.
> > Secondly, the writing of the second para looks weaker than the rest of the paper. There is probably a minor typo (...still evaluate "if" to increase...). The para seems to talk about data subsets as a choice of fidelity, however, the last 2 sentences are a bit difficult to parse and be sure that I understand correctly.
> > Does it mean that if $r$ is set to be a small number and not enough rungs are created or epochs trained, PASHA cannot do its crisscrossing measure and therefore does not find good epsilons and basically does not offer the speedups? However, if we take the number of updates or the like as resource levels, then we allow PASHA to formulate more rungs and calculate epsilon more robustly.

---

> > > ### Author Response · Authors · 2022-11-23
> > > **Detailed follow-up response (1/2)**
> > >
> > > We provide detailed answers to the specific comments that needed further clarifications.
> > >
> > > A-1) The accuracy reported is the maximum accuracy if we retrain for 200 epochs and use early stopping (more specifically checkpoint selection - it is good practice to do that for neural networks). Apologies for any confusion in this. We have mentioned in the paper too that it is the maximum accuracy (best accuracy) - in the first paragraph of section 5.2.
> > >
> > > A-2) Thanks for the suggestion, we can certainly use that as the random baseline instead. We have run these experiments and the results are as follows. Comparing these results with the ones in the paper, we see the random baselines with 256 and 2560 samples give almost identical results.
> > >
> > > CIFAR-10
> > > | Approach | Accuracy (%) | Runtime | Speedup factor | Max resources |
> > > |--|--|--|--|--|
> > > | Random baseline | 72.88 $\pm$ 19.20 | 0.0h $\pm$ 0.0h | NA | 0.0 $\pm$ 0.0 |
> > >
> > > CIFAR-100
> > > | Approach | Accuracy (%) | Runtime | Speedup factor | Max resources |
> > > |--|--|--|--|--|
> > > | Random baseline  | 42.83 $\pm$ 18.20 | 0.0h $\pm$ 0.0h | NA | 0.0 $\pm$ 0.0 |
> > >
> > > ImageNet16-120
> > > | Approach | Accuracy (%) | Runtime | Speedup factor | Max resources |
> > > |--|--|--|--|--|
> > > | Random baseline  | 20.75 $\pm$ 9.97 | 0.0h $\pm$ 0.0h | NA | 0.0 $\pm$ 0.0 |
> > >
> > > WMT
> > > | Approach | Accuracy (%) | Runtime | Speedup factor | Max resources |
> > > |--|--|--|--|--|
> > > | Random baseline  | 33.93 $\pm$ 21.96 | 0.0h $\pm$ 0.0h | NA | 0.0 $\pm$ 0.0 |
> > >
> > > ImageNet
> > > | Approach | Accuracy (%) | Runtime | Speedup factor | Max resources |
> > > |--|--|--|--|--|
> > > | Random baseline  | 36.94 $\pm$ 31.05 | 0.0h $\pm$ 0.0h | NA | 0.0 $\pm$ 0.0 |
> > >
> > > A-3) 5-epoch baseline often gives good prediction quality, but not always and is (significantly) slower than PASHA.
> > >
> > > B-5,6 and C-1,2) $c_c$ is going to be evaluated until 8 epochs, but in the example it was only trained up to 6 epochs so far, so that is what is used for the current calculation. Yes, it is in set $S$ because it is a candidate promoted to the top rung - note that in the example shown we assume all these pairs already belong to the set $S$ i.e. there was criss-crossing within each pair $(c_a, c_b)$, $(c_a, c_c)$ and $(c_b, c_c)$ (we will improve the final version of the paper by explicitly mentioning this assumption). The criss-crossing can happen in any epoch, including the ones from previous rungs - but we only consider configurations that are in the latest rung and take the scores from their latest available epoch.
> > >
> > > C-5) Thanks for the additional clarifications to the question, we have now tried to do an analysis experiment that gives us insights into this. In particular, we do the following: 1) take the configurations that share the top soft rank at the end of HPO (last rung reached by PASHA), 2) find the average across the retraining accuracies for all these configurations, 3) compare this with retraining accuracy of the best configuration at the end of HPO (argmax at the last rung reached by PASHA).
> > >
> > > The results show that in some cases we can indeed use a randomly selected configuration from the top soft rank and obtain strong performance, potentially even better one. However, using argmax overall appears to give more stable results (especially for the more challenging cases) and hence would be the preferred option. We note that argmax remains the most intuitive option to use because we are expected to return only one configuration to the user. Of course, the results also suggest there is the possibility to improve over argmax selection (even if it is unclear how), which we are happy to mention as potential future work in the conclusion.
> > >
> > > CIFAR-10
> > >
> > > | Approach | Accuracy (%) |
> > > | ------------- | ------------------|
> > > | ASHA  | 93.85 $\pm$ 0.25 |
> > > | PASHA argmax | 93.57 $\pm$ 0.75 |
> > > | PASHA top soft rank | 94.10 $\pm$ 0.20 |
> > >
> > > CIFAR-100
> > > | Approach | Accuracy (%) |
> > > | ------------- | ------------------|
> > > | ASHA |  71.69 $\pm$ 1.05 |
> > > | PASHA argmax |  71.84 $\pm$ 1.41 |
> > > | PASHA top soft rank | 71.99 $\pm$ 0.88 |
> > >
> > > ImageNet16-120
> > >
> > > | Approach | Accuracy (%) |
> > > | ------------- | ------------------|
> > > | ASHA | 45.63 $\pm$ 0.81 |
> > > | PASHA argmax | 45.13 $\pm$ 1.51 |
> > > | PASHA top soft rank | 43.01 $\pm$ 2.34 |

---

> > > ### Author Response · Authors · 2022-11-23
> > > **Detailed follow-up response (2/2)**
> > >
> > > D-1) Yes, if the rankings do not change, it is a sign they are stable and we can stop HPO if we have already reached the stopping condition of the number of sampled configurations. If we have not yet sampled the maximum number of configurations, then we sample more of them and train them up to the current maximum resources. If the rankings are still stable, then we stop HPO. This is desirable because it allows us to obtain the speedups - it is not necessary to obtain a more robust estimate because we have judged the estimate to be already sufficiently robust, allowing us to save resources. If we increased the resources even if the ranking is stable, there would be no speedups.
> > >
> > > OC-5) We go towards ASHA and explore higher fidelities if the rankings are not stable because in such cases it is necessary to obtain a more robust estimate to decide which configuration is the best. However, apart from that PASHA would sample more candidates than ASHA if for example given total budget as the stopping criterion - this could allow it to find better candidates.
> > >
> > > OC-6) We use GPs in the same way as MOBSTER, so we learn one GP across all fidelities (note that MOBSTER in Syne Tune is the original implementation of the authors). Additional details to answer the questions - using Syne-Tune documentation: *MOBSTER is choosing a new configuration by maximizing the expected improvement (EI) acquisition function at a certain resource level `r_acq`. Since we are ultimately interested in performance at `r = max_t`, we would like to set `r_acq = max_t` as early as possible. On the other hand, EI may not be reliable at a resource level if too little metric data has been observed there (i.e., too few trials reached this level). MOBSTER is setting `r_acq` to the largest rung level `r <= max_t` for which at least `resource_acq_bohb_threshold` (= 3) metric values have been recorded.*
> > >
> > > Fig 3) Yes, the y-axis (value of epsilon) is measured in terms of accuracy, so it indeed converges to about 5%. x-axis is the number of updates to state information as you summarised.
> > >
> > > Fig 4) We could certainly visualise the LCs only up to the resources selected by PASHA - but also the current analysis shows that criss-crossing happens (including the early stages) and that we can use it to automatically estimate the value of epsilon. Yes, the plot also intends to show that the three configurations are very similar and any of them could be chosen as the final one.
> > >
> > > Section 6) When we talk about the number of data points processed, we mean the number of non-unique examples seen so far - so not the dataset size, but the sum of all mini batch sizes so far (hence proportional to the number of iterations). We’re happy to improve the writing in this section in the final revision. To explain the final part of section 6: if initial resources $r$ represent a large part of full training (e.g. 1/3), there are not enough rounds of promotion to see significant speed-up from PASHA because we need multiple rounds of pruning to see benefits of stopping the pruning early. However, this can be resolved by redefining minimum resources $r$ appropriately, as explained in the paper.
> > >
> > > Thanks for following up on these detailed comments as well, we hope we have been able to clarify them as needed.

---

> > ### Comment · Reviewer_tjkb · 2022-11-20
> > **Overall response to the rebuttal**
> >
> > Thank you for the clarifications and revisions to the paper. A lot of the confusion or misunderstanding has been cleared indeed. I must reiterate that the work is interesting and intuitive simplicity is always a nice feature to have and I do like the work.
> > However, there are two things I am not able to resolve when it comes to an update to the score.
> >
> > * Imagine a scenario where I want to tune a model and need to choose an HPO algorithm. If I have parallel resources and enough time to run ASHA, I do not see why I should not do that given I am more likely to get a configuration likely to be evaluated or compared to a configuration trained till convergence. Offering a scale or degree of improvement. For early stopping methods in the literature, they rely either on meta-learning to estimate performance *at convergence* or estimate performance by extrapolating some parametric form of the shape of an LC. PASHA is not doing either but relying on configurations it has explored so far to conclude that estimates at $r<R$ are consistent at $R$. This is akin to saying Successive Halving can choose to not evaluate the one configuration at $R$ but select the top-1 from the rung at $R/\eta$. Could certainly work in practice. However, when PASHA is being pitched as a general HPO and NAS algorithm, I am struggling to see how I trust this information knowing PASHA has no idea of actual performances at anything higher. This links to the potential PASHA pathology I mentioned earlier wherein PASHA may never increase resources if I run it indefinitely. If I do not have enough budget and want to quickly get a good configuration, PASHA appears to be a good choice. However, I guess I am from the following school of thought where I would be more likely to use an algorithm that I know will not have a potential failure mode for a new problem.
> >
> >
> > * Given the admittance that PASHA is an extension of ASHA and thus carries with it the failure modes of ASHA (not all multi-fidelity methods share these), the extent of contribution becomes confusing. PASHA can become an impressive go-to algorithm. However, given its inherited failure modes and the point above, if PASHA is an add-on one could use on top of ASHA to gain speedups, the contribution feels lesser than it is or can be.
> >
> >
> > **TL;DR**
> > As much as I like the work, I am a bit perplexed about why the design of PASHA will not, over time, allow for exploring higher resources. Since that will still offer speedups over ASHA and potentially provide performance benefits. Moreover, offer a guarantee to solve any new problem with either immense speed-ups over ASHA or being worst-case bound by ASHA. That would be an awesome feature to run ASHA with that leaves no reservation.
> >
> > If you can engage me in this thought and help allay my concerns, I shall definitely look to increase my score.
> > Thanks!

---

> > > ### Author Response · Authors · 2022-11-23
> > > **Follow-up response**
> > >
> > > Thank you very much for the additional questions and giving us the opportunity to clarify the further few points. We are glad to hear that most of the previous questions and misunderstandings have already been resolved by our responses.
> > >
> > > **General HPO selection:** If the user has enough resources and time to run ASHA, they can certainly use ASHA instead of PASHA. In fact, if they can afford to do random search or full-scale BO, those methods are preferable as they are safer/more robust than ASHA and other multi-fidelity methods.
> > >
> > > However, we know there are many ML practitioners for whom these methods are too expensive, in terms of time and computation (which directly translates to costs), e.g. because they have very large datasets or simply because of limited time or money. Normally they would have to resort to simple baselines such as the one-epoch baseline evaluated in the paper, which carries far higher risks. For ML practitioners who use the cloud to run HPO, it is very beneficial if the HPO costs them e.g. only \\$1,000 rather than \$15,000 if it can give them a similarly good configuration. More broadly we lower the entrance barrier for new HPO users, which is important for the AutoML community too.
> > >
> > > PASHA allows for exploring higher resources if it sees it would be beneficial, so the comments about PASHA not exploring higher resources may be a misunderstanding. To explain more, we note that the fact that PASHA does not train for more epochs if the ranking is stable (and samples more configurations instead) is a feature, not a failure mode. If the ranking of configurations is stable, then the algorithm sees that resources would be better spent trying additional new configurations rather than evaluating the configurations for longer. There would be no resource savings if PASHA did not have any mechanism to restrict the exploration of higher rungs (right now it explores those only to the extent needed).
> > >
> > > Overall, our method was tested extensively, we also ran additional experiments and they all confirmed our initial statements. The method is cheap, effective and it can serve a number of practical use-cases. It also prevents users from setting the wrong maximum budget $R$, which is not a problem for AutoML researchers using standard benchmarks but it is far from trivial in practice.
> > >
> > > **PASHA as an ASHA extension:** PASHA builds on ASHA and multi-fidelity methods more broadly, but these extensions are novel and non-trivial (more fundamentally we can see most of current ML research published at top conferences as extensions one way or another). In the worst case possible, PASHA defaults to ASHA (which happens if the rankings are consistently unstable). The benefits brought by PASHA are non-incremental as they allow scaling HPO to qualitatively different problems, enabling us to do HPO where only very limited approaches were previously possible.
> > >
> > > In terms of failure modes, we agree that PASHA has the same failure modes as ASHA, and we recognize that methods such as HyperBand are able to alleviate these failure modes. However, HyperBand is significantly more costly to run than ASHA, and the focus of our work is on efficiency and scalability of HPO. ASHA is a widely used method because in most cases its failure modes do not represent an issue. PASHA could be extended to an analogous HyperBand version, but this is not desirable because our primary goal is to make HPO significantly more scalable than currently possible. We have tested our method extensively, and we have shown it gives strong results.
> > >
> > > We believe we have been able to address these additional questions, and we hope the reviewer is now willing to recommend acceptance for the paper.

---

> > > > ### Comment · Reviewer_tjkb · 2022-11-24
> > > > **Final request and increased score**
> > > >
> > > > Thanks a lot for engaging in this discussion with the patiently elaborated details. Appreciate it!
> > > >
> > > > Overall, the paper is much much clearer after this discussion. However, as you might understand, I *needed* this discussion on top of the paper, which suggests that the paper's writing can be improved or made clearer.
> > > > To be more constructive, I think the positioning of the algorithm and its exact scope of contribution should be cleared out more explicitly. I am totally for practical tools and that is what PASHA brings to the table. Can I use PASHA to get SOTA by tuning my new model? Cannot be sure. However, PASHA could give me something better than random in a time that no other competitor seemingly can. That could be called out much more explicitly.
> > > > Naturally, the paper could clear out the contribution of the paper and possible future work by mentioning that PASHA inherits everything ASHA. Barring a data-driven heuristic to limit or choose the max rung/resource. But the other limitations are shared too. This is mentioned in a couple of places in the paper but it may be a bit disparate for the reader.
> > > >
> > > > Secondly, I still think the primary contribution is as the authors note, "reducing the entry barrier to HPO for DL", and again, that should be called out more explicitly. Since PASHA modifies ASHA ever so slightly to kind of *warmstart* ASHA (if one could draw that image). The user can accordingly choose to continue with HPO for longer while in parallel proceed with work with a good working setting found by PASHA. I unfortunately still struggle to see PASHA's solution being as reliable or more competitive than the "classical" methods with guarantees. That doesn't take away PASHA's ability to be extremely efficient in practice. Something that is quite essential in modern scenario.
> > > >
> > > > Thirdly, regarding the behaviour of PASHA, the assumptions on the learning curve properties, etc. it would be great to have some analysis in the appendix with single worker runs as it makes the algorithm a bit more interpretable than the parallel setting. For example, does the stability of soft ranking depend on the rung width? Can adding newer configurations affect epsilon and soft ranking such that they end up adding a new resource when a rung keeps getting wider? All in all, as simple and effective as the idea is, it does take some thinking to wrap one's head around why it is works and is expected to work in any problem.
> > > >
> > > > Also, for the algorithm and paper to be implemented easily, some more details could be provided:
> > > > * What are the step sizes over which epochs are selected for epsilon computation?
> > > > * Does PASHA require prediction over the validation set at every epoch then? Does the *runtime* in the table capture this evaluation overhead?
> > > > * Or is PASHA flexible enough to work with any epoch-step size (probably should be <= $\eta$ as I understand) and therefore can save on evaluation overhead for every epoch? (if so, why not mention it as a flexible PASHA feature)
> > > > * If the *max* performance over an LC is returned as performance, that means it accounts for possible divergence. In that case, is the epsilon still calculated at the latest epoch recorded for a config (where performance might have dropped, also affecting noise range)?
> > > > * Some explanations on the high variance of maximum resource used by PASHA (in the tables)
> > > >
> > > > **Increased score**
> > > >
> > > > The paper addresses an important requirement in HPO and offers a nice step towards democratizing HPO for large models. The authors have been excellent with their detailed responses and some of the paper revisions have helped too. The clarification of epsilon computation using *epochs* and not *rungs* was quite crucial in understanding the work. I personally would of course like to see more updates to the paper as I have listed above, in order for the paper to read better and the algorithm to be easier to understand, implement, and adopt in practice. I would encourage the authors to consider the points above.
> > > >
> > > > In view of the rather nice discussion, I have increased the score to recommend acceptance. It is a work that should be published for more viewership, scrutiny, stress testing, and improvement.

---

> > > > > ### Author Response · Authors · 2022-11-24
> > > > > **Thank you**
> > > > >
> > > > > Thank you very much for increasing the score, we really appreciate it. We are glad to hear you believe the paper should be published - and we will improve the final version of the paper based on your feedback. We have enjoyed the in-depth discussion with you and are grateful for your effort and time spent on it - it is very valuable for improving the paper and ultimately making the paper more impactful.

---

### Author Response · Authors · 2022-11-16
**Author response - new version of the paper, more experiments and extended analysis**

Many thanks to all reviewers for the valuable suggestions on how to improve our paper. We have added several experiments and analyses to the appendix, as recommended in some of the reviews. The key additional new results include the following:
* Additional baselines (Table 7 and 8): two, three and five epoch baselines. The results show that with such simple approaches, the performance is still not sufficiently close to ASHA and in some cases these are even slower than PASHA (because they do not do any pruning). The experiments with a five-epoch baseline highlight the speed of PASHA. These experiments also further highlight that with such simple baselines it is difficult to obtain robust results because sometimes e.g. 1 or 2 epochs give better results than 3 or 5 epochs.
* Investigation of percentile value $N$ for calculating epsilon (Table 9): the results show a limited sensitivity in the tops of the scale and other values from the top range working well.
* Investigation of how the value of epsilon evolves (Figure 3): this analysis makes it clear that our approach can be inspected (is not black-box) and its behaviour is interpretable - we can see what is the epsilon value used in soft ranking.
* Analysis of learning curves (Figure 4 and 5): we show that for similarly performing configurations, there is indeed random noise present, allowing us to calculate a sensible value for the epsilon parameter.
* Additional analysis on PD1 (Table 13): now with alternative ranking functions, showing that the other options work well too (but were not selected because we want to keep our method simple and easy to use).

In addition, we have also improved the writing in the paper (all changes are in red). We believe these updates make the paper stronger, clarify all questions and weaknesses (in combination with our replies). We will be happy to add further clarifications or details if needed.

---

### Comment · Area_Chair_n5Wb · 2022-12-11
**Choice of benchmarks?**

Dear authors,

As the AC for this paper, in my discussion with the reviewers the choice of benchmarks came up, and I would like to give you an opportunity to comment on this in the remainder of the available time in the rebuttal phase.

There are many tabular benchmarks available next to NAS bench 201 and PD1 (e.g. other NAS benchmarks, LC-Bench, FCNet, HPO-Bench, TaskSet, YAHPO gym, learning curves by Mohr & van Rijn, etc). With PASHA being purely heuristically motivated, the question is whether using two benchmarks (with 3 and 2 datasets, respectively) yields a comprehensive enough view on its performance. Compute cost should not be an issue, since these are tabular benchmarks and PASHA doesn't have a model, so experiments on a new benchmark should run through very quickly. Your experiments section appears to mostly aim to demonstrate that your algorithm works well („In this section we empirically quantify the advantage provided by PASHA.“) rather than to study its pros and cons. You state that you left out the other PD1 datasets since PASHA would not show any performance improvements. But would there also be similarly high speedups without loss of accuracy in other benchmarks, or did you select the benchmarks for the paper where PASHA yields the best performance? Especially since LC-Bench and FCNet are natively supported in SyneTune (see https://openreview.net/forum?id=BVeGJ-THIg9), it is surprising that you did not evaluate on them. Would you like to comment on this?

Best,
AC

---

> ### Author Response · Authors · 2022-12-12
> **Response (1/2)**
>
> Thanks for giving us the opportunity to clarify the choice of the benchmarks. Public benchmarks for HPO/NAS generally set the unit of measure for resources to one epoch because the maximum amount of resources allocated to a configuration is large (e.g., 100, 200) and going sub-epoch does not make a significant difference. In our case, the focus is on identifying the best configuration with as little resources as possible. For this purpose we would like to measure resources with higher granularity (e.g.,100 gradient updates) since this will condition the scheduling of our rungs and will provide the algorithm with more opportunities to stop the optimization process. Unfortunately, to the best of our knowledge, no public benchmark provides such granularity.
>
> To reconcile the need for reproducibility and standardisation of the experiments and the need for an environment where the algorithm could make a non-trivial decision, we decided to run our test on benchmarks with a larger number of epochs. In fact, PASHA needs at least two rungs before deciding to stop. If the maximum amount of resources that the benchmark allows for a configuration barely exceeds the two rungs, the speedup will be negligible. If the resources barely exceed three rungs, it will have only one opportunity to stop the optimization early and so on.
>
> This is not a problem that is observed in practice since it is generally possible to set sub-epoch resource granularity. This limitation is due to the construction of the most common benchmarks for HPO/NAS and their choice to fix the minimum resource level to 1 epoch. With this we do not intend to look down on the previous work done by the creators of those benchmarks, for which we are *very* grateful, but just to explain the origin of these limitations. We hope this discussion provides useful insights into the choices we made.

---

> ### Author Response · Authors · 2022-12-12
> **Response (2/2)**
>
> In order to answer your request and extend the information already provided in Section 6, we ran experiments on LCBench. This benchmark limits the maximum amount of resources per configuration to 50 epochs, so when using $\eta=3$ and setting the minimum resource level to 1 epoch, it is a challenging testbed for an algorithm like PASHA.
> Overall, the results confirm an accuracy level on-par with ASHA. While, as expected, the speedup is reduced compared to the experiments on NASBench, in several cases PASHA achieves a 20+% speedup with peaks around 40%.
>
> The table below reports accuracies for both ASHA and PASHA and the time speedup provided by PASHA over ASHA.
>
> | Dataset | ASHA accuracy (%) | PASHA accuracy (%) | PASHA speedup |
> | ----------- | ----------- | ----------- | ----------- |
> | APSFailure | 97.52 $\pm$ 0.92 | 97.01 $\pm$ 0.75 | 1.3x |
> | Amazon_employee_access | 94.01 $\pm$ 0.40 | 94.21 $\pm$ 0.00 | 1.1x |
> | Australian | 83.35 $\pm$ 0.33 | 83.35 $\pm$ 0.51 | 1.1x |
> | Fashion-MNIST | 86.70 $\pm$ 1.87 | 86.34 $\pm$ 1.32 | 1.1x |
> | KDDCup09_appetency | 98.22 $\pm$ 0.00 | 98.22 $\pm$ 0.00 | 1.1x |
> | MiniBooNE | 86.13 $\pm$ 1.57 | 86.24 $\pm$ 1.62 | 1.4x |
> | adult | 79.14 $\pm$ 0.85 | 79.14 $\pm$ 0.85 | 1.2x |
> | airlines | 59.57 $\pm$ 1.32 | 59.22 $\pm$ 0.76 | 1.4x |
> | albert | 64.31 $\pm$ 0.99 | 64.23 $\pm$ 0.61 | 1.2x |
> | bank-marketing | 88.34 $\pm$ 0.07 | 88.38 $\pm$ 0.00 | 1.2x |
> | blood-transfusion-service-center | 79.92 $\pm$ 0.20 | 76.99 $\pm$ 6.00 | 1.1x |
> | car | 86.60 $\pm$ 6.41 | 86.60 $\pm$ 6.41 | 1.1x |
> | christine | 71.05 $\pm$ 1.17 | 70.15 $\pm$ 1.85 | 1.2x |
> | cnae-9 | 94.10 $\pm$ 0.31 | 94.44 $\pm$ 0.11 | 1.0x |
> | connect-4 | 62.28 $\pm$ 6.87 | 65.69 $\pm$ 0.04 | 1.2x |
> | covertype | 59.76 $\pm$ 3.24 | 61.64 $\pm$ 1.64 | 1.2x |
> | credit-g | 70.30 $\pm$ 0.84 | 70.79 $\pm$ 0.68 | 1.1x |
> | dionis | 64.58 $\pm$ 9.89 | 64.58 $\pm$ 9.89 | 1.1x |
> | fabert | 56.11 $\pm$ 10.89 | 53.47 $\pm$ 9.75 | 1.1x |
> | helena | 19.16 $\pm$ 3.20 | 19.16 $\pm$ 3.20 | 1.1x |
> | higgs | 66.48 $\pm$ 3.16 | 66.48 $\pm$ 3.16 | 1.1x |
> | jannis | 58.92 $\pm$ 2.38 | 59.63 $\pm$ 2.81 | 1.4x |
> | jasmine | 75.85 $\pm$ 0.36 | 75.55 $\pm$ 0.68 | 1.0x |
> | jungle_chess_2pcs_raw_endgame_complete | 72.86 $\pm$ 4.69 | 74.94 $\pm$ 7.84 | 1.3x |
> | kc1 | 80.32 $\pm$ 4.37 | 80.86 $\pm$ 3.37 | 1.2x |
> | kr-vs-kp | 92.50 $\pm$ 3.93 | 90.95 $\pm$ 4.70 | 1.0x |
> | mfeat-factors | 98.21 $\pm$ 0.15 | 98.15 $\pm$ 0.15 | 1.1x |
> | nomao | 94.12 $\pm$ 0.60 | 94.25 $\pm$ 0.64 | 1.1x |
> | numerai28.6 | 52.03 $\pm$ 0.55 | 52.30 $\pm$ 0.24 | 1.3x |
> | phoneme | 76.65 $\pm$ 2.65 | 75.42 $\pm$ 2.87 | 1.1x |
> | segment | 83.15 $\pm$ 2.54 | 83.15 $\pm$ 2.54 | 1.0x |
> | sylvine | 90.57 $\pm$ 1.87 | 90.89 $\pm$ 2.04 | 1.0x |
> | vehicle | 71.76 $\pm$ 2.57 | 71.76 $\pm$ 2.57 | 1.1x |
> | volkert | 50.72 $\pm$ 1.91 | 50.72 $\pm$ 1.91 | 1.1x |
>
>
> We thank you for suggesting this experiment and we hope that these results can give you additional insights in the performance of PASHA. We plan to integrate these findings in the paper to give readers a better understanding of the overall performance of the algorithm.
>
> If you have additional requests or feedback we will do our best to answer them.

---

### Author Response · Authors · 2023-02-28
**How to use PASHA to obtain large speedups**

PASHA is designed to give large speedups compared to ASHA when there are many rounds of promotion - e.g. when there are 200 epochs and rung levels 1, 3, 9, 27, etc. (the case in NASBench201). If there are e.g. 50 epochs and the same rung levels, the speedup is expected to be smaller, which can be seen in our investigation with variable maximum resources (Appendix E) and also the additional LCBench results (Appendix D and the discussion here). We recommend having a maximum amount of resources at least 100 times larger than the minimum amount of resources when using $\eta = 3$ (default). If the number of epochs is small, it is best to redefine the resources in terms of iterations (gradient updates), to enable large speedups even in these cases (as discussed in Section 6 of the paper and other places).

---

### Decision · Program_Chairs · 2023-01-20

**Decision:**

Accept: poster

**Justification For Why Not Higher Score:**

The authors only showed results on hand-selected benchmarks where their method performs well and did not show the (known) cases where it does not lead to large speedups / yields worse results.

**Justification For Why Not Lower Score:**

The method is intuitive and simple and yields good results when learning curves don't cross much (which is the case in many but not all benchmarks).

**Metareview: Summary, Strengths And Weaknesses:**

This paper introduces PASHA, a simple heuristic resource allocation method that avoids allocating higher resources to compare sets of configurations whose ranking has stabilized with lower amounts of resources. After a lot of interaction between reviewers and authors, during which the authors added substantial new evidence, all reviewers increased their scores to lie above the acceptance threshold. A key concern in the decision process was the current setup of the experiments section, which are designed to show the strong performance of PASHA in a horse race on a few benchmarks, hand-picked as those where PASHA works well: 2 datasets from PD1 (with the authors specifically acknowledging that PASHA wouldn't work well on other PD1 datasets) and the 3 datasets from NAS bench 201. There are a lot of other benchmarks available (e.g. other NAS benchmarks, LC-Bench, FCNet, HPO-Bench, TaskSet, YAHPO gym, learning curves by Mohr & van Rijn, etc), and a better goal for the experiments would have been a comprehensive evaluation that scientifically studies cases where PASHA works well and cases where it does not work well. This is computationally cheap since all of these benchmarks are surrogate benchmarks. Asked about this, the authors added experiments on LC-Bench on short notice that do indeed show much more modest speedups (only around 20%).
Overall, I recommend acceptance due to the method's simplicity, intuitiveness, and good results in some cases, which can make this a very useful building block of future HPO algorithms. It is not at all clear, though, whether this approach by itself constitutes a robust HPO method. I strongly encourage the authors to add more comprehensive experiments (including any negative results), to scientifically study the benefits and disadvantages of the method.

**Note From Pc:**

if the above contains the word "oral" or "spotlight" please see: "oral" presentation means -> notable-top-5% and "spotlight" means -> notable-top-25%. As stated in our emails, we are disassociating presentation type from AC recommendations